# CAUSALITY ≠ INVARIANCE:
# FUNCTION AND CONCEPT VECTORS IN LLMS

**Gustaw Opiełka**[*]          **Hannes Rosenbusch**          **Claire E. Stevenson**

*University of Amsterdam*

## ABSTRACT

Do large language models (LLMs) represent concepts abstractly, i.e., independent of input format? We revisit Function Vectors ($\mathcal{FV}$s), compact representations of in-context learning (ICL) tasks that causally drive task performance. Across multiple LLMs, we show that $\mathcal{FV}$s are not fully invariant: $\mathcal{FV}$s of the same concept are nearly orthogonal when extracted from different input formats (e.g., open-ended vs. multiple-choice). We identify Concept Vectors ($\mathcal{CV}$s), which carry more stable concept representations. Like $\mathcal{FV}$s, $\mathcal{CV}$s are composed of attention head outputs; however, unlike $\mathcal{FV}$s, the constituent heads are selected using Representational Similarity Analysis (RSA) based on whether they encode concepts consistently across input formats. While these heads emerge in similar layers to $\mathcal{FV}$-related heads, the two sets are largely distinct, suggesting different underlying mechanisms. Steering experiments reveal that $\mathcal{FV}$s excel in-distribution, when extraction and application formats match (e.g., both open-ended in English), while $\mathcal{CV}$s generalize better out-of-distribution across both question types (open-ended vs. multiple-choice) and languages. Our results show that LLMs do contain abstract concept representations, but these differ from those that drive ICL performance.

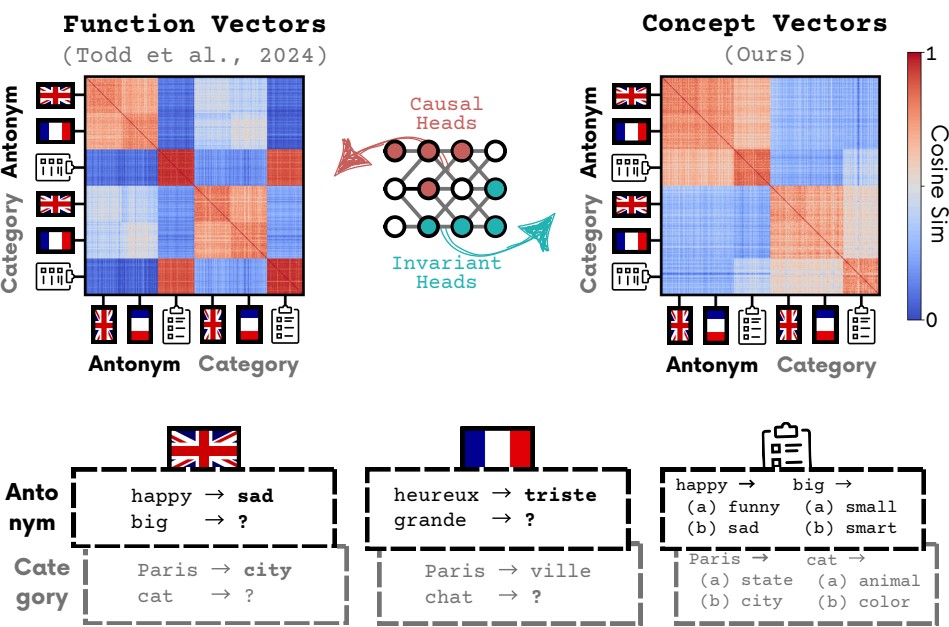

Figure 1: *Function vs. Concept Vectors*. **Top**: Similarity matrices for $\mathcal{FV}$s (left) and $\mathcal{CV}$s (right) in Llama 3.1 70B; cells show how similar two prompt representations are (warmer = more similar). **Middle**: Schematic highlighting the distinction between heads with causal effect (Activation Patching-selected) and heads that encode format-invariant structure (RSA-selected). **Bottom**: Example prompts for two concepts across three formats (EN open-ended, FR open-ended, multiple-choice). **Takeaway**: $\mathcal{FV}$s cluster by input format; $\mathcal{CV}$s cluster by concept across formats.

---

[*]Corresponding author: `g.j.opielka@uva.nl`

# 1 INTRODUCTION

Do large language models represent concepts abstractly, i.e., in a way that is stable across surface form? We focus on *relational concepts*: mappings between entities, such as linking a word to its antonym. Cognitive science has long argued that abstract representation of such structure underlies human generalization (Gentner, 1983; Hofstadter, 1995; Mitchell, 2020). This capability allows identifying, by analogy, that "hot → cold" and "big → small" share the same oppositional relation, independent of the specific words or how the task is presented. Recent work shows that LLMs exhibit representational structures similar to humans (Pinier et al., 2025; Du et al., 2025; Doerig et al., 2025), raising the question: *do the abstract representations hypothesized to support analogical reasoning actually drive LLM performance on such tasks?*

We find that LLMs do contain abstract relational concept information, but the components that encode it differ from those that causally drive in-context learning (ICL) behavior. This separation challenges the *single-circuit hypothesis* that format-invariant representations are what primarily enable ICL.

We revisit Function Vectors ($\mathcal{FV}$s)—compact vectors formed by summing outputs of a small set of attention heads that mediate ICL (Todd et al., 2024; Hendel et al., 2023; Yin & Steinhardt, 2025). Because $\mathcal{FV}$s transfer across contexts (e.g., differently formatted prompts and natural text), they are often treated as encoding the underlying concept (Zheng et al., 2024; Griffiths et al., 2025; Bakalova et al., 2025; Brumley et al., 2024; Fu, 2025). We update this view: $\mathcal{FV}$s are not fully invariant. For the same concept, $\mathcal{FV}$s extracted from different input formats (open-ended vs. multiple-choice) are nearly orthogonal, indicating that $\mathcal{FV}$s mix concept with format (§2.2.1).

To isolate format-invariant structure, we contrast *activation patching* (AP), which localizes components with causal effects on outputs, with *representational similarity analysis* (RSA) (Kriegeskorte, 2008), which localizes components whose representations organize by concept independent of format. Using RSA to select heads and then summing their activations yields Concept Vectors ($\mathcal{CV}$s). Across seven relational concepts, three input formats (open-ended English, open-ended French, multiple-choice), and four models (Llama 3.1 8B/70B; Qwen 2.5 7B/72B), we find that $\mathcal{CV}$ heads arise in similar layers but are largely disjoint from $\mathcal{FV}$ heads, suggesting separable mechanisms for invariance vs. causality (§2.2.2).

Finally, we test whether $\mathcal{CV}$s can steer. In steering experiments, $\mathcal{FV}$s produce larger in-distribution gains when extraction and application formats match (§3.2.1), whereas $\mathcal{CV}$s generalize more consistently out-of-distribution across question type and language (§3.2.2) and produce fewer format artifacts (e.g., tokens and language from extraction prompts; §3.2.3).

Overall, our contributions are as follows:

- $\mathcal{FV}$**s are not input-invariant.** They mix relational concepts with input format; same-concept $\mathcal{FV}$s differ sharply across formats.

- **RSA reveals** $\mathcal{CV}$ **heads.** These heads encode relational concepts at a higher level of abstraction than $\mathcal{FV}$ heads.[1]

- $\mathcal{CV}$ **and** $\mathcal{FV}$ **heads are disjoint.** $\mathcal{FV}$s and $\mathcal{CV}$s are realized by different attention heads, suggesting that abstract concept representations are distinct from the mechanisms that causally drive ICL performance.

- **Steering trade-off.** $\mathcal{FV}$s steer more strongly in-distribution, while $\mathcal{CV}$s generalize more consistently out-of-distribution, albeit with smaller absolute gains.

# 2 IN SEARCH OF INVARIANCE

We test whether concept representations are stable across surface form, using AP (causal heads) and RSA (format-invariant heads) across models, datasets, and formats. We then form Function/Concept Vectors to compare clustering by format vs. concept; AP/RSA heads lie in similar layers but show minimal top-K overlap.

---

[1]We expand on what we mean by "higher level of abstraction" in the Discussion (§5).

## 2.1 METHODS

### 2.1.1 MODELS

We test Llama 3.1 (8B, 70B) and Qwen 2.5 (7B, 72B) models (Meta AI, 2024; Qwen et al., 2025). All models are autoregressive, residual-based transformers (Vaswani et al., 2023). Each model, $f$ internally comprises of $\mathcal{L}$ layers. Each layer is composed of a multi-layer perceptron (MLP) and $J$ attention heads $a_{\ell j}$ which together produce the vector representation of the last token of layer $\ell$, $\mathbf{h}_\ell = \mathbf{h}_{\ell-1} + \text{MLP}_\ell + \sum_{j \in J} a_{\ell j}$ (Elhage et al., 2021).

### 2.1.2 TASKS

**Datasets** We define a dataset as one concept expressed in one input format (e.g., Antonym in open-ended English). For each dataset we build a set of in-context prompts $P_d = \{p_d^i\}$ where $i$ indexes individual prompts within dataset $d$. Each prompt contains few-shot input–output examples $(x, y)$ that illustrate the same concept, followed by a query input $x_q^i$ whose target output $y_q^i$ is withheld. The input–output pairs $(x, y)$ were either sourced from prior work or generated using OpenAI's GPT-4o (see Appendix D for details). Example prompts are provided in Appendix A.

**Concepts.** We consider seven concepts:

- **Antonym**   Map a word to one with opposite meaning (e.g., hot → cold).
- **Categorical**   Map a word to its semantic category (e.g., apple → fruit).
- **Causal**   Map a cause to an effect (e.g., rain → wet).
- **Synonym**   Map a word to one with similar meaning (e.g., big → large).
- **Translation**   Translate a word to another language (e.g., house → maison).
- **Present–Past**   Convert a verb from present to past tense (e.g., run → ran).
- **Singular–Plural**   Convert a noun from singular to plural (e.g., cat → cats).

**Input formats.** We vary only the prompt's surface format; the $(x, y)$ relation stays the same. Formats:

- Open-ended ICL in English (`OE-EN`)
- Open-ended ICL in a different language (French or Spanish; `OE-FR` or `OE-ES`)
- Multiple-choice ICL in English (`MC`)

We use 5-shot prompts for open-ended and 3-shot for multiple-choice to reduce computational load given prompt length. Altogether, we have 21 datasets (7 concepts × 3 input formats). We build 50 prompts per dataset (total $N = 1050$ prompts).

### 2.1.3 ACTIVATION PATCHING

Activation patching replaces specific activations with cached ones from a *clean* run to assess their impact on the model's output. The cached activations are then inserted into selected model components in a *corrupted* run, where the systematic relationships in the prompt are disrupted. For example, in an antonym ICL task, consider a *clean prompt*: Hot → **Cold**, Big → **Small**, Clean → **?** and a *corrupted prompt*: House → Cold, Eagle → Small, Clean → **?** The goal of activation patching is then to localize model components that push the model to the correct answer, **Dirty**, on the corrupted prompt.

We compute the *causal indirect effect* (CIE) for each attention head $a_{\ell j}$ as the difference between the probability of predicting the expected token $y$ when processing the corrupted prompt $\tilde{p}$ with and without the transplanted mean activation $\bar{\mathbf{a}}_{\ell j}$ from clean runs:

$$\text{CIE}\big(a_{\ell j}\big) = f\Big(\tilde{p} \mid \mathbf{a}_{\ell j} := \bar{\mathbf{a}}_{\ell j}\Big)[y] - f(\tilde{p})[y] \tag{1}$$

We then compute the *average indirect effect* (AIE) over a collection $\mathcal{D}$ of all datasets (§2.1.2). [2]

$$\text{AIE}(a_{\ell j}) = \frac{1}{|\mathcal{D}|} \sum_{d \in \mathcal{D}} \frac{1}{|\tilde{\mathcal{P}}_d|} \sum_{\tilde{p}_i \in \tilde{\mathcal{P}}_d} \text{CIE}\big(a_{\ell j}\big) \tag{2}$$

where $\tilde{\mathcal{P}}_d$ denotes the set of corrupted prompts for dataset $d$.

---

[2]*Note*: Unlike Todd et al. (2024) we compute AIE scores across all input formats, not `OE-ENG` only.

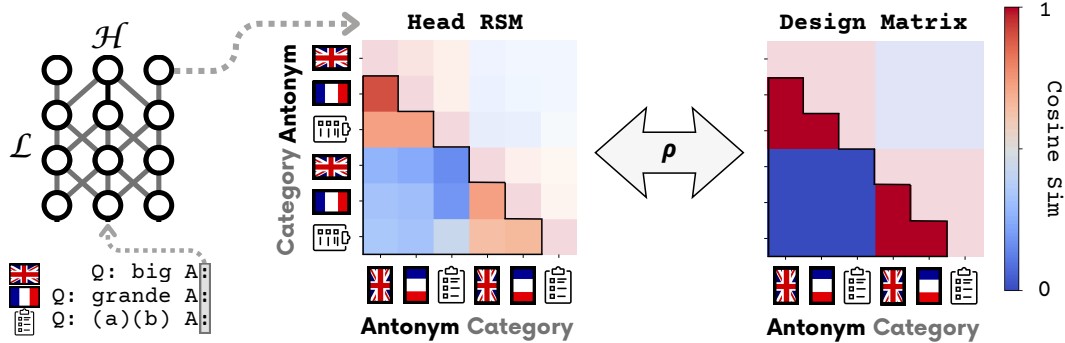

Figure 2: *Representational Similarity Analysis (RSA)*. For each attention head, we compute a representational similarity matrix (RSM) over prompts spanning concepts and input formats (cosine similarity of head outputs). We construct a binary design matrix that marks pairs sharing the same concept, independent of format. The RSA score for a head is Spearman's $\rho$ between the lower-triangular entries of the RSM and the design matrix; higher $\rho$ indicates stronger concept-invariant encoding.

### 2.1.4 REPRESENTATIONAL SIMILARITY ANALYSIS

To find attention heads encoding concepts invariant to input formats, we employ representational similarity analysis (RSA; Kriegeskorte (2008)).

For each attention head $a_{\ell j}$ we compute representational similarity matrices (RSMs) where $v_i$ denotes the output extracted from $a_{\ell j}$ for the $i$th prompt $p_i \in P_N$, and $\theta(\cdot, \cdot)$ is a cosine similarity function.

$$
\text{RSM} = \begin{bmatrix} 1 & \cdots & \theta(v_1, v_N) \\ \vdots & \ddots & \vdots \\ \theta(v_N, v_1) & \cdots & 1 \end{bmatrix}
\tag{3}
$$

We then construct a binary design matrix, DM, where each entry is set to 1 if the corresponding pair of prompts share the same attribute value, and 0 otherwise. In this paper, we consider two attributes: (1) `concept` - does a pair of prompts illustrate the same concept, regardless of the input format? and (2) `prompt_format` - does a pair of prompts have the same question type (i.e. open-ended or multiple-choice)?

We then quantify the alignment between the RSM and DM for the lower-triangles (since similarity matrices are symmetric) using the non-parametric Spearman's rank correlation coefficient ($\rho$).

To localize attention heads carrying invariant concept information we compute the RSA for each attention head obtaining a single Concept RSA score for each attention head.

$$
\text{Concept-RSA}\big(a_{\ell j}\big) = \rho(\text{RSM}_{\ell j}, \text{Concept-DM})
\tag{4}
$$

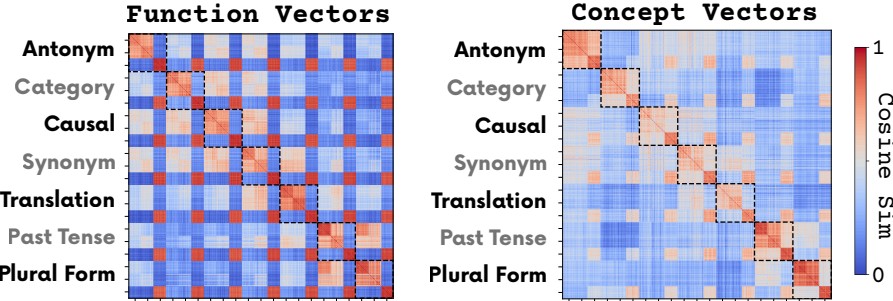

Figure 3: *Similarity matrices*. Full similarity matrices extracted from top $K = 5$ heads in $\mathcal{CV}$s and $\mathcal{FV}$s in Llama 3.1 70B for all concepts. See Appendix B for other models.

### 2.1.5 FUNCTION & CONCEPT VECTORS

To form Function/Concept Vectors we create sets of top $K$ ranking attention heads, $\mathcal{A}_{\mathcal{FV}}$ and $\mathcal{A}_{\mathcal{CV}}$, based on their AIE and RSA scores respectively. Function/Concept Vectors for prompt $i$ are then computed as the sum of activations for this prompt, $\mathbf{a}_{\ell j}^i$, from the sets $\mathcal{A}_{\mathcal{FV}}$ and $\mathcal{A}_{\mathcal{CV}}$ respectively.

$$\mathcal{FV}_i = \sum_{a_{\ell j}^i \in \mathcal{A}_{\mathcal{FV}}} \mathbf{a}_{\ell j}^i \quad \mathcal{CV}_i = \sum_{a_{\ell j}^i \in \mathcal{A}_{\mathcal{CV}}} \mathbf{a}_{\ell j}^i \tag{5}$$

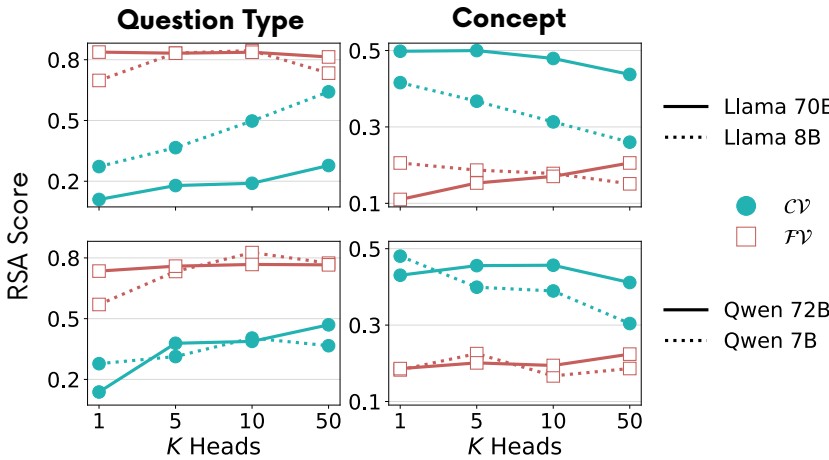

Figure 4: *Concept vs. format RSA*. Question type and Concept RSA scores for $\mathcal{CV}$s and $\mathcal{FV}$s in all models. **Takeaway**: $\mathcal{CV}$s encode more concept information and less input format than $\mathcal{FV}$s.

## 2.2 RESULTS

### 2.2.1 CONCEPT VECTORS ARE MORE INVARIANT TO INPUT FORMAT

We test invariance to input format by computing RSA with design matrices for concept and question type (following the setup in §2.1.4). We form $\mathcal{FV}$s/$\mathcal{CV}$s by summing the top-$K$ heads ranked by AIE/RSA (Eq. 5). Across models and $K$, $\mathcal{CV}$s show higher concept RSA and lower question-type RSA than $\mathcal{FV}$s (Figure 4), indicating that $\mathcal{FV}$s encode format more strongly while $\mathcal{CV}$s track concept. Consistently, similarity matrices for Llama 3.1 70B cluster by concept across formats for $\mathcal{CV}$s, but by format for $\mathcal{FV}$s (Figure 3), where within-format type $\mathcal{FV}$ clusters are nearly identical with mean cosine similarity $= 0.90$. $\mathcal{CV}$s nonetheless exhibit a weaker within-format type cluster (mean cosine similarity $= 0.55$), suggesting they retain some low-level format information. Overall, however, $\mathcal{CV}$s remain markedly more invariant to input format than $\mathcal{FV}$s.

| Model | K=3 | K=5 | K=10 | K=20 | K=50 | K=100 |
|---|---|---|---|---|---|---|
| Llama-3.1 8B | 0 | 0 | 1 | 1 | **12** | **28** |
| Llama-3.1 70B | 0 | 0 | 0 | 0 | **1** | **6** |
| Qwen2.5 7B | 0 | 0 | 0 | 4 | **15** | **39** |
| Qwen2.5 72B | 0 | 0 | 0 | 1 | **3** | **13** |

Table 1: *RSA–AIE head overlap*. Overlap between RSA and AIE heads (number of overlapping heads among top-$K$). Bold numbers indicate overlap significantly above chance ($p < 0.05$; details in Appendix E). **Takeaway**: $\mathcal{FV}$s and $\mathcal{CV}$s are composed of different attention heads.

### 2.2.2 FUNCTION & CONCEPT VECTORS ARE COMPOSED OF DIFFERENT ATTENTION HEADS

If we compare which heads are selected by the two procedures, we see that $\mathcal{FV}$s and $\mathcal{CV}$s are composed of different attention heads. First, we ranked each head for each method, i.e., AIE (§2.1.3)

for $\mathcal{FV}$s and by Concept-RSA (§2.1.4) for $\mathcal{CV}$s. Then we examined depth and top-$K$ overlap. Layer-averaged scores show similar layer profiles (Figure 5), but head identities barely overlap: for $K \leq 20$ the intersection is near zero and stays small at larger $K$ (Table 1). We also note that AIE scores are highly sparse: their histogram peaks at zero with a long right tail (Figure 12)—so only a few heads have measurable causal effect. Together this supports that AIE-selected *causal* heads are largely distinct from the *invariant*, RSA-selected heads.

To ensure this separation is not an artifact of patching within the same format, we also performed cross-format activation patching (e.g., extracting activations from open-ended prompts and patching them into multiple-choice). This procedure continued to identify the same FV heads and did not identify CV heads (see Appendix O), confirming that FVs are the primary causal drivers regardless of input format.

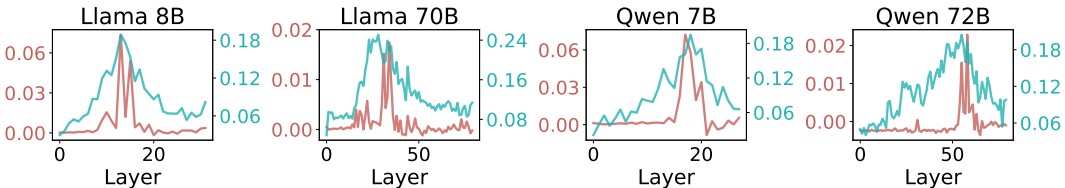

Figure 5: *Layer-wise AIE vs. RSA.* AIE and RSA scores averaged across all heads per layer. **Takeaway**: $\mathcal{FV}$ and $\mathcal{CV}$ heads are in similar layers.

## 3  CAN CONCEPT VECTORS STEER?

We now test whether these invariant heads can steer: we introduce how we construct vectors, the AmbiguousICL setup with conflicting cues, and the intervention protocol. $\mathcal{FV}$s win in-distribution; $\mathcal{CV}$s transfer better out-of-distribution with fewer format artifacts, at a cost of smaller gains.

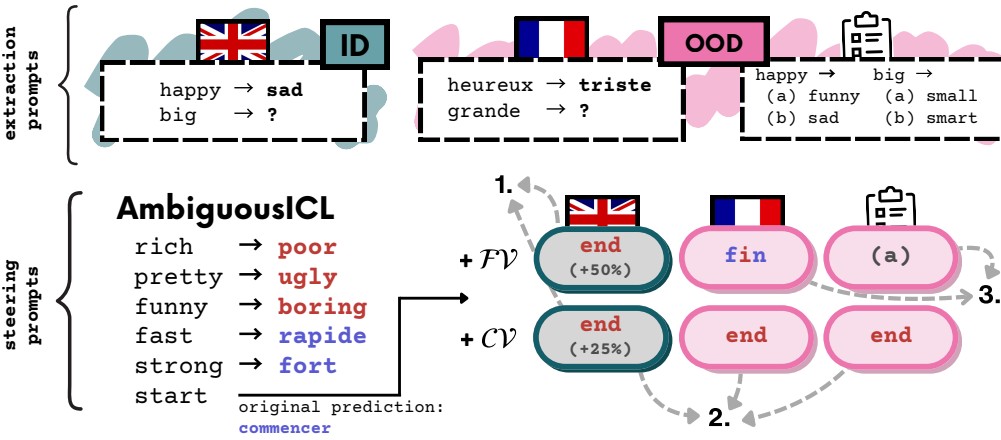

Figure 6: *Overview of steering results.* **Top**: We extract $\mathcal{CV}$s and $\mathcal{FV}$s from antonym ICL prompts in formats that are in-distribution (ID; `OE-ENG`) or out-of-distribution (OOD; `OE-FR`, `MC`) relative to the AmbiguousICL task (bottom-left). **Bottom-left**: We interleave two concepts—**antonym** and **EN→FR translation**—within one prompt; the model's original prediction is the French translation. **Bottom-right**: Predictions after steering. **Takeaways**: (1) $\mathcal{FV}$s yield larger ID gains. (2) $\mathcal{CV}$s show more stable OOD effects across formats. (3) $\mathcal{FV}$s can conflate concept with input format (e.g., French version of antonym and multiple-choice formatting).

### 3.1  STEERING METHODS

**Steering Vectors Construction.** For each concept and input format (`OE-ENG`, `OE-FR`, `MC`), we compute for every selected head $a_{\ell j}$ the mean last-token activation across the 50 *extraction prompts*

of that concept–format. We then form one vector per format by summing these mean activations over the top-$K$ heads selected for $\mathcal{CV}$ or $\mathcal{FV}$ (as in Eq. 5, but using per-format means in place of per-prompt activations). This yields one ID vector (`OE-ENG`) and two OOD vectors (`OE-FR`, `MC`) per concept.

**AmbiguousICL Task.** We evaluate on AmbiguousICL tasks (Figure 6): each prompt interleaves two concepts (3 then 2 exemplars) followed by a query. The second concept is always English→French translation. Unsteered models tend to continue with the second concept; we aim to steer toward the first. Note that steering prompts are distinct from the extraction prompts used to construct the vectors. This setup is diagnostic: it tests whether representations encode abstract relational structure independent of extraction prompts' surface format. To perform well in this setup requires consistency between ID and OOD performance.

**Steering with $\mathcal{CV}$s and $\mathcal{FV}$s.** We add a vector $\mathbf{v}$ to the last-token residual stream at a chosen layer:

$$\mathbf{h}_\ell \leftarrow \mathbf{h}_\ell + \alpha\mathbf{v} \tag{6}$$

We measure effectiveness as $\Delta P = P_{\text{after}}(y) - P_{\text{before}}(y)$, averaged over 100 prompts per concept (see Figure 22 for Top-1 accuracy). We sweep $\alpha$ and $K$ and report the best per model (Appendix F).

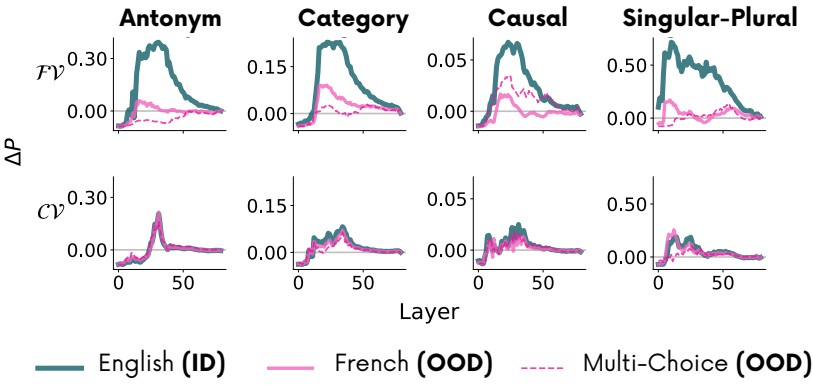

Figure 7: *Steering effect across layers.* We inject $\mathcal{CV}$s and $\mathcal{FV}$s into Llama-3.1-70B and plot the change in target-token probability ($\Delta P$) for four representative concepts (columns). Curves compare ID extraction format with OOD formats relative to the AmbiguousICL task (Figure 6). Higher $\Delta P$ means the model assigns more probability to the expected token than the unsteered model. **Takeaways**: (1) $\mathcal{FV}$s typically achieve larger ID gains but often drop OOD. (2) $\mathcal{CV}$s yield smaller gains yet show more stable OOD behavior across formats. See Figure 16 for other concepts/models.

## 3.2 STEERING RESULTS

### 3.2.1 FUNCTION VECTORS OUTPERFORM CONCEPT VECTORS IN DISTRIBUTION

Extracted from `OE-ENG` (ID setting), $\mathcal{FV}$s yield the largest gains on ambiguous prompts (Figure 7). $\mathcal{CV}$s also help but with smaller $\Delta P$ and minimal zeroshot effect (Figure 17). At the token level both vectors lift plausible English antonyms in the ID case (Table 2).

### 3.2.2 CONCEPT VECTORS ARE MORE STABLE OUT OF DISTRIBUTION

**Performance gains ($\Delta P$).** Out of distribution (extracting vectors from `OE-FR` or `MC`), $\mathcal{CV}$s more often maintain positive effects across formats, whereas $\mathcal{FV}$s frequently degrade—especially for `MC`—and only occasionally stay consistent for specific concepts/models (Figs. 7, 16). $\mathcal{CV}$s raise the probability of the correct English answer across formats, and their top-$\Delta$ tokens remain concept-aligned (Table 2). Crucially, the key finding is not absolute performance but consistency: $\mathcal{CV}$s increase the probability of producing similar concept-aligned tokens regardless of extraction format.

**Distributional consistency (KL).** To quantify consistency across formats independent of absolute gains, we compare the model's next-token distributions after steering with ID and OOD vectors. For

**Query:** `salty →`

|  | + Antonym | Top Δ Tokens |
|---|---|---|
| $\mathcal{FV}$ | OE-ENG | `_sweet` (+56%), `_fresh` (+16%), `_bland` (+6%), `_taste` (+3%), `_uns` (+2%) |
|  | OE-FR | `_su` (+31%), `_dou` (+27%), `_frais` (+5%), `_fade` (+5%), `_ins` (+3%) |
|  | MC | `_(` (+53%), `_A` (+1%), `_\n` (+1%), `_space` (+0%), `_)` (+0%) |
| $\mathcal{CV}$ | OE-ENG | `_sweet` (+49%), `_fresh` (+8%), `_bland` (+3%), `_taste` (+3%), `_uns` (+3%) |
|  | OE-FR | `_sweet` (+54%), `_fresh` (+9%), `_bland` (+3%), `_uns` (+3%), `_taste` (+2%) |
|  | MC | `_sweet` (+35%), `_fresh` (+12%), `_bland` (+4%), `_uns` (+3%), `_taste` (+3%) |

Table 2: *Token-level steering effects.* Top tokens with largest probability gains when injecting $\mathcal{CV}$s or $\mathcal{FV}$s into Llama-3.1-70B on the AmbiguousICL prompt (query shown above). Results shown at the layer with the strongest in-distribution effect per vector. Without intervention, the model predicts French translation `_sa` (from *salé*) with 49%; antonym `_sweet` has 2%. English antonyms in red, French in blue, and the opening bracket (MC token) in green.

each concept and vector type, we select the top 5 layers that achieve the highest ID $\Delta P$. At each selected layer we compute KL divergence

$$D_{KL}\big[p(\mathbf{x}\,|\,\mathbf{v}_{\text{OOD}})\,\|\,p(\mathbf{x}\,|\,\mathbf{v}_{\text{ID}})\big]$$

between the post-intervention distributions at the query token, where lower values indicate more similar effects of ID and OOD vectors. We average this KL divergence over prompts and selected layers to obtain one score per concept, and then summarize per model (Figure 8). Across models, $\mathcal{CV}$s yield lower KL than $\mathcal{FV}$s. The CV–FV KL gap is larger for `MC` than for `OE-FR`.

### 3.2.3 FUNCTION VECTORS MIX CONCEPT WITH INPUT FORMAT

Out of distribution, $\mathcal{FV}$s reflect both prompt format and concept. When vectors are extracted from `OE-FR`, they push the model toward the French translation of the concept (e.g., French antonyms), and when extracted from `MC`, they increase the probability of format tokens such as the opening bracket (Table 2). We quantify the language effect by measuring $\Delta P$ for the French translation across concepts (Figure 13). In the larger models, $\mathcal{FV}$s substantially increase the probability of the French token, whereas $\mathcal{CV}$s remain near zero; in smaller models the effect is negligible. Notably, $\mathcal{FV}$s extracted from open-ended Spanish prompts induce almost the same bias toward the French translation as $\mathcal{FV}$s extracted from French prompts (Figure 14), even though the AmbiguousICL alternatives are French only. This pattern suggests that $\mathcal{FV}$s capture a generic translation/foreign-language signal tied to the extraction format rather than language-specific content. Combined with the MC bracket effect (Figure 15), these findings indicate that $\mathcal{FV}$s mix concept with surface format, while $\mathcal{CV}$s are comparatively format-invariant.

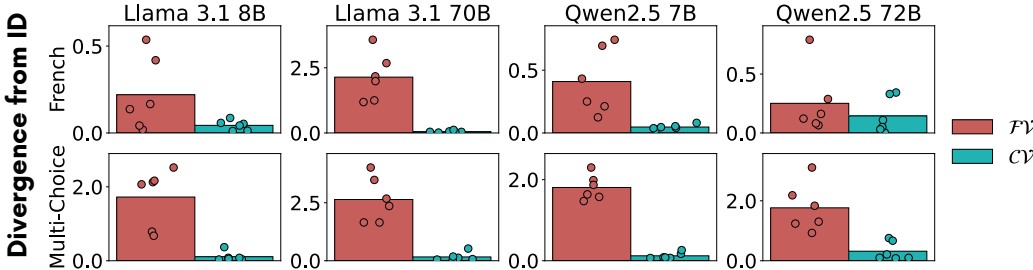

Figure 8: *Consistency of steering effects.* KL divergence between the probability distributions after the models were steered with an ID (OE-ENG) and OOD vectors (OE-FR [top], MC [bottom]). Lower values indicate more similar effects of ID and OOD vectors. **Takeaway**: $\mathcal{CV}$s steer the models more consistently than $\mathcal{FV}$s.

## 4  RELATED WORK

**Attention Head Categorization.** Recent work has made significant progress in characterizing specialized attention heads that process in-context learning (ICL) tasks. For instance, Olsson et al. (2022) identified induction-heads, which Yin & Steinhardt (2025) found can develop into $\mathcal{FV}$-heads during training. Other specialized head types include semantic-induction heads (Ren et al., 2024), symbol-abstraction heads (Yang et al., 2025), and various others (Zheng et al., 2024). Our work extends this line of research by identifying $\mathcal{CV}$ heads, attention heads that invariantly represent concepts in ICL tasks at high levels of abstraction.

**Linear Representation of Concepts.** A substantial body of research has established that concepts are represented linearly in LLMs' representational space (Mikolov et al., 2013; Arora et al., 2016; Elhage et al., 2022). This phenomenon, often termed the "Linear Representation Hypothesis" (Park et al., 2024), has been extensively studied across various tasks and domains. Hernandez et al. (2024) demonstrated that relational concepts—similar to those we study in this paper—can be decoded from LLM activations using linear approximation. Subsequent work by Merullo et al. (2025) revealed that the success of such decoding depends on the frequency of concepts in the pretraining corpora, which may explain why some concepts are represented more consistently than others in our study. Our findings contribute to this literature in two ways: (1) providing further support for the Linear Representation Hypothesis, and (2) extending previous work on relational concept representations by localizing specific attention heads that carry such representations and demonstrating their invariance to input formats.

**Symbolic-like reasoning in LLMs.** Recent work has demonstrated that LLMs can exhibit symbol-like representational properties even without explicit symbolic architecture (Feng & Steinhardt, 2024; Yang et al., 2025; Griffiths et al., 2025). Yang et al. (2025) define symbolic processing as requiring two key properties: (1) invariance to content variations, and (2) indirection through pointers rather than direct content storage. Our $\mathcal{CV}$s exhibit both properties: they are invariant to input format changes and function as pointers to content stored elsewhere, unlike $\mathcal{FV}$s which directly store content (§3.2.3).

## 5  DISCUSSION

Our results separate two representational roles in LLMs: components that *cause* strong ICL performance and components that *encode* abstract concept structure. Function Vectors ($\mathcal{FV}$s) occupy the first role, steering models effectively when extraction and application formats match, but deteriorating out of distribution (formats/languages). Conversely, Concept Vectors ($\mathcal{CV}$s) built from RSA-selected heads encode concepts at a *higher level of abstraction* and generalize more robustly across languages and question types, albeit with smaller causal effects. This supports a view that invariance and causality are mediated by largely distinct mechanisms in similar layers.

**Layers of abstraction.** We define abstraction as encoding relational structure (e.g., "antonym") while discarding surface details (e.g., "English, multiple-choice"). Our results suggest $\mathcal{FV}$s do capture abstract task information: they reliably encode concepts within a format (Figure 19) and are causally effective even across formats (Appendix O). However, their orthogonality across formats and retention of surface signals (e.g., MC brackets) reveal that they conflate this abstract content with surface form. In contrast, $\mathcal{CV}$s cluster by concept regardless of format. Thus, $\mathcal{FV}$s operate at a lower level of abstraction ("antonym in MC format"), while $\mathcal{CV}$s operate at a higher level ("antonym"), independent of surface form.

**Relation to Function Vectors.** Prior work shows that $\mathcal{FV}$s compactly mediate ICL and can transfer across contexts (Todd et al., 2024). We refine this: $\mathcal{FV}$ portability is strong within families of prompts, but is not fully invariant to surface format. Same-concept $\mathcal{FV}$s extracted from different formats are nearly orthogonal and can carry language/format signals (e.g., French subword or multiple-choice bracket tokens), while $\mathcal{CV}$s track concept across formats with less surface content. This distinction can be framed as *equivariance* vs. *invariance*: $\mathcal{FV}$s adapt to extraction format (e.g., producing French antonyms from French prompts, MC formatting tokens from MC prompts), whereas $\mathcal{CV}$s steer toward similar outputs regardless of format. Finally, we do not propose $\mathcal{CV}$s as competitors to $\mathcal{FV}$s, but rather highlight a mechanistic dissociation: $\mathcal{FV}$s drive behavior (causality) while $\mathcal{CV}$s represent abstract structure (invariance).

These findings have implications for theoretical models of ICL, such as recent work by Bu et al. (2025) which posits the retrieval of a single function vector $a^f$ for a function $f$. Our results suggest this model is incomplete: given the orthogonality of $\mathcal{FV}$s across formats, the function vector is better conceptualized as format-conditional $a(f, \phi)$, implying convergence to multiple format-specific basins rather than a single global minimum. Furthermore, we find that $\mathcal{FV}$s and $\mathcal{CV}$s are orthogonal (even within the same format) which suggests that task representations partition into distinct abstract and format-specific subspaces, rather than residing in a single unified space.

**Implications for steering and interpretability.** The dissociation between $\mathcal{FV}$s and $\mathcal{CV}$s suggests a practical trade-off. For *maximal in-distribution control*, $\mathcal{FV}$s are preferable. For *robust out-of-distribution control* or probing abstract knowledge, $\mathcal{CV}$s are more reliable.

However, $\mathcal{CV}$s appear to require the concept to be already present in the prompt to exert influence. In zero-shot steering (Figure 17) and activation patching—which require inducing or restoring a task "from scratch"—$\mathcal{CV}$s are ineffective. In contrast, in AmbiguousICL, where the concept is present but competing, $\mathcal{CV}$s successfully steer by amplifying the existing abstract signal. Thus, $\mathcal{FV}$s seem necessary to *instantiate* a task, while $\mathcal{CV}$s can *modulate* it once present.

Methodologically, AP identifies what causally drives behavior, while RSA reveals how representations organize by concept. This distinction highlights that behavioral control and abstract representation can be mediated by different mechanisms.

**Analogies and abstract representation.** Hill et al. (2019) propose that "analogies are something like the functions of the mind": concepts achieve their flexibility by being represented abstractly enough to permit context-dependent adaptation across diverse domains of application. This view predicts that relational concepts like *antonym* or *causation* should function identically whether presented as open-ended prompts, multiple-choice questions, or in different languages. Our findings offer a mechanistic refinement: while LLMs do form abstract relational representations ($\mathcal{CV}$s), these are largely distinct from the components that causally drive ICL behavior ($\mathcal{FV}$s). This dissociation suggests that analogical task performance may not require—or primarily rely on—the most abstract conceptual representations. Instead, LLMs appear to solve ICL tasks via more format-specific mechanisms, even though they do form abstract representations.

**Limitations and Future Directions.** Our $\mathcal{CV}$ head selection targeted heads that encode *all* concepts simultaneously; this global criterion may miss concept-specific heads, which a per-concept RSA could reveal. We also did not probe how $\mathcal{FV}$s and $\mathcal{CV}$s emerge during model training or how they interact during inference.

We propose two possible hypotheses that could be explored in future work:

1. **CVs and FVs interact during inference as detection/execution mechanisms.** Previous work by Lindsey et al. (2025) has discovered model features that seem to fire just before the model produces a certain type of output (e.g., a "capital" feature that fires just before the model outputs a name of a capital), and ones that fire more generally when the text mentions different capitals. Other work found that ICL tasks can be understood from an "encoder/decoder" perspective (Han et al., 2025), where the encoder encodes the task into a latent space and the decoder decodes the latent space into the output. Both of these findings suggest that models separate the task encoding and execution into distinct mechanisms which can be linked to $\mathcal{CV}$s (encoding/detection) and $\mathcal{FV}$s (execution).

2. **CVs and FVs do not interact during inference; CVs are simply a backup circuit.** Another possibility is that the two mechanisms are independent. Other works have found *backup circuits* where models can form multiple, partially redundant circuits and compensatory self-repair under ablations (McGrath et al., 2023; Wang et al., 2022). Given that a) both sets of heads are in similar layers (Figure 5), suggesting $\mathcal{CV}$s and $\mathcal{FV}$s may operate in parallel or via lateral information flow within the residual stream, rather than strict deep-hierarchical dependencies and b) that $\mathcal{CV}$ heads do not seem to have causal effects in usual ICL tasks (since they were not identified by AP), this hypothesis is also plausible. What is more Davidson et al. (2025) found that different prompting methods yield a different causal tasks representations, therefore it is possible that ICL in LLMs consist of multiple, separate mechanisms.

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

# A PROMPT EXAMPLES

## A.1 OPEN-ENDED (5-SHOT)

```
Q: resistant
A: susceptible

Q: classify
A: disorganize

Q: posterior
A: anterior

Q: goofy
A: serious

Q: stationary
A: moving

Q: hairy
A:
```

## A.2 MULTIPLE-CHOICE (3-SHOT)

```
Instruction: Q: unveil A: ?
(a) optional
(b) mild
(c) con
(d) conceal
Response: (d)

Instruction: Q: hooked A: ?
(a) unhooked
(b) stale
(c) sturdy
(d) sell
Response: (a)

Instruction: Q: spherical A: ?
(a) unconstitutional
(b) flat
(c) demand
(d) healthy
Response: (b)

Instruction: Q: minute A: ?
(a) conservative
(b) hour
(c) retail
(d) awake
Response: (
```

# B SIMILARITY MATRICES FOR OTHER MODELS

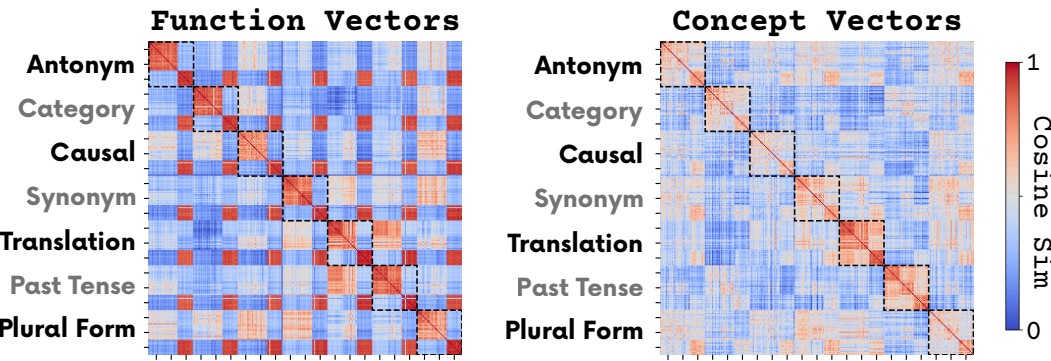

Figure 9: Similarity matrices extracted from top $K = 1$ heads in $\mathcal{CV}$s and $\mathcal{FV}$s in Llama 3.1 8B.

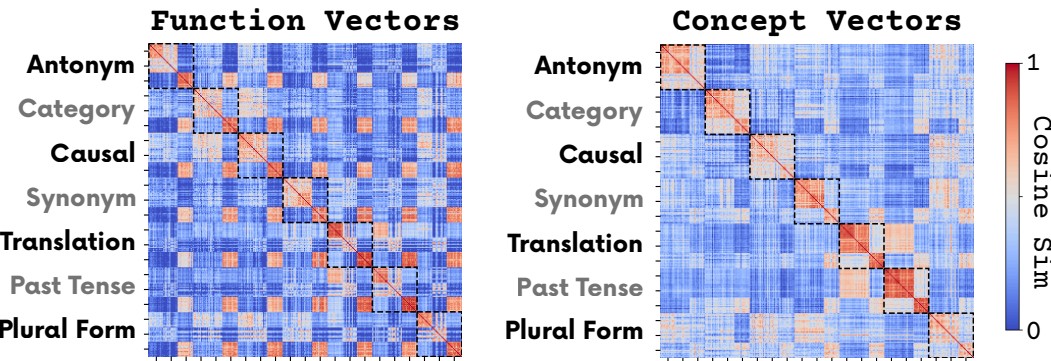

Figure 10: Similarity matrices extracted from top $K = 1$ heads in $\mathcal{CV}$s and $\mathcal{FV}$s in Qwen 2.5 7B.

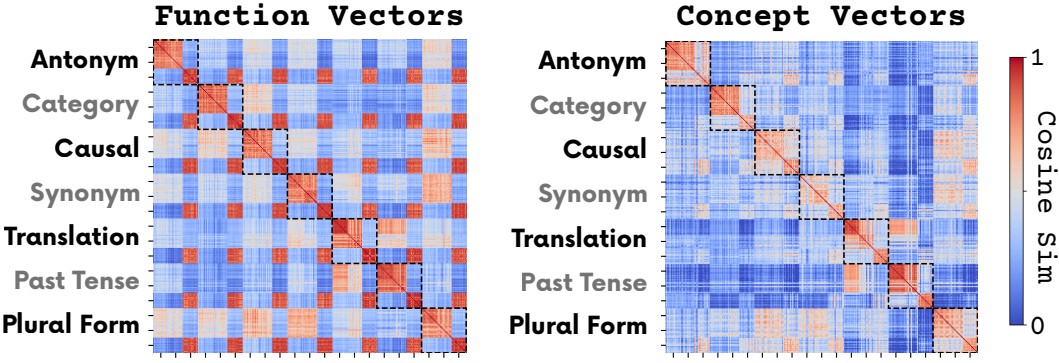

Figure 11: Similarity matrices extracted from top $K = 2$ heads in $\mathcal{CV}$s and $\mathcal{FV}$s in Qwen 2.5 72B.

## C  AIE Scores

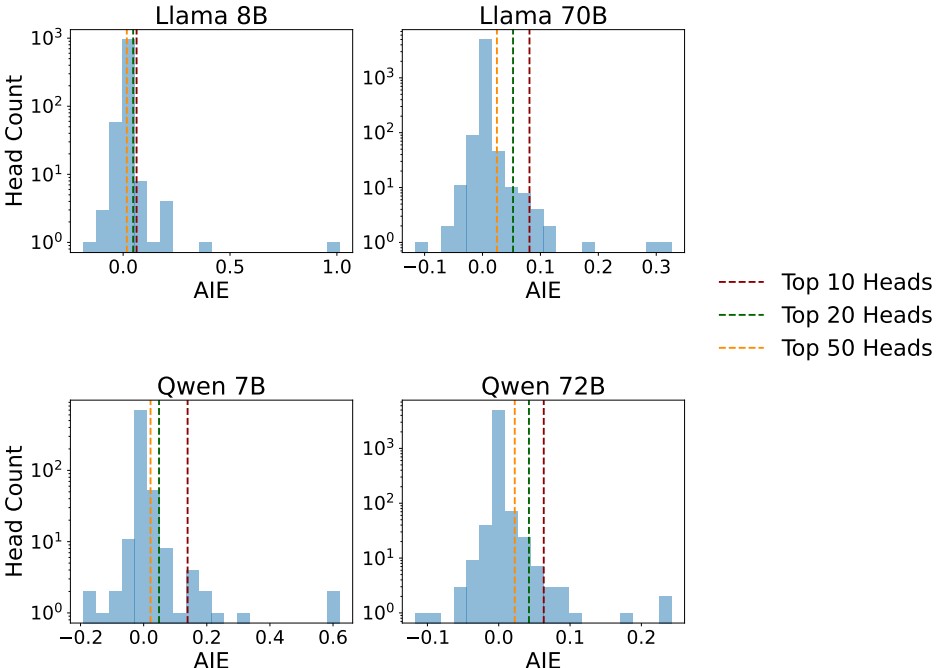

Figure 12: Histogram of AIE scores for Llama 3.1 8B, 70B, Qwen 2.5 7B, and Qwen 2.5 72B. Note, the y-axis is on a log-scale. **Takeaway**: AIE scores are highly sparse.

## D  Data Generation Process

**Concept sourcing**: For most concepts (antonym, synonym, translation, present–past, singular–plural), we sourced word pairs from the datasets used by Todd et al. (2024). For categorical and causal concepts, we generated word pairs using OpenAI's GPT-4o model (OpenAI, 2024).

**Translation generation**: French and Spanish translations were created using DeepL's translation service (DeepL SE, 2025) to ensure high-quality, contextually appropriate translations.

**Generated concepts (categorical and causal)**: We prompted GPT-4o to generate exemplar:category pairs (e.g., "apple:fruit", "blue:colour") and cause:effect pairs (e.g., "stumble:fall", "storm:flood"). The model was given examples of the desired format and asked to produce 100 pairs per batch. We generated pairs in batches of 100 until reaching approximately 1000 examples per concept, with retry mechanisms to ensure sufficient coverage. The final datasets were saved as JSON files containing input-output pairs.

**Quality filtering**: Generated pairs underwent several filtering steps: (1) removal of duplicates based on input words, (2) exclusion of pairs containing underscores or numbers, (3) restriction to single words or two-word phrases (maximum one space per input/output), and (4) conversion to lowercase for consistency.

**Multiple choice format**: For multiple choice prompts, we generated four options per question by randomly sampling three additional outputs from the same concept dataset, ensuring all four options were unique. The correct answer was randomly positioned among the four options.

# E  SIGNIFICANCE TEST FOR RSA–AIE HEAD OVERLAP

We assess whether the observed overlap between the top-$K$ heads selected by Concept-RSA and by AIE is larger than expected by chance under a simple null model. Let $N$ denote the total number of attention heads in the model (layers $\times$ heads per layer). For a fixed $K$, each method selects a size-$K$ subset of heads. Under the null hypothesis that these two subsets are independent, uniformly random size-$K$ subsets of $\{1, \ldots, N\}$, the overlap size

$$X = |S_{\text{RSA},K} \cap S_{\text{AIE},K}|$$

follows a hypergeometric distribution $X \sim \text{Hypergeom}(N, K, K)$.

For an observed intersection $x$, we report the one-sided tail probability

$$p_{\geq x} = \Pr\left[X \geq x\right] = \sum_{t=x}^{K} \frac{\binom{K}{t}\binom{N-K}{K-t}}{\binom{N}{K}} \, .$$

Entries with $p_{\geq x} < 0.05$ are typeset in bold in Table 1.

# F  STEERING HYPERPARAMETERS

To optimize the intervention performance, we conduct a hyperparameter search for two parameters:

- $\alpha$: the steering weight that controls the strength of the intervention
- $K$: the number of attention heads to extract for concept vector computation

We evaluate the following parameter ranges:

- $K \in \{1, 3, 5, 10, 20, 50\}$ for the number of heads
- $\alpha \in \{1, 3, 5, 10, 15\}$ for the steering weight

The hyperparameter optimization is performed separately for each model using antonym prompts. We select the parameter combination that maximizes the average steering effect across all input formats. This ensures that our chosen hyperparameters generalize well across different prompt structures. We report the best hyperparameters for each model in Table 3.

| Model | Best $K$ | Best $\alpha$ |
|---|---|---|
| Llama 3.1 8B | 1 | 10 |
| Llama 3.1 70B | 5 | 10 |
| Qwen 2.5 7B | 3 | 10 |
| Qwen 2.5 72B | 5 | 15 |

Table 3: Optimal hyperparameters for steering interventions across different models. $K$ represents the number of attention heads used for $\mathcal{FV}$/$\mathcal{CV}$ extraction, while $\alpha$ controls the intervention strength.

# G   INPUT FORMAT MIXING IN FUNCTION VECTORS

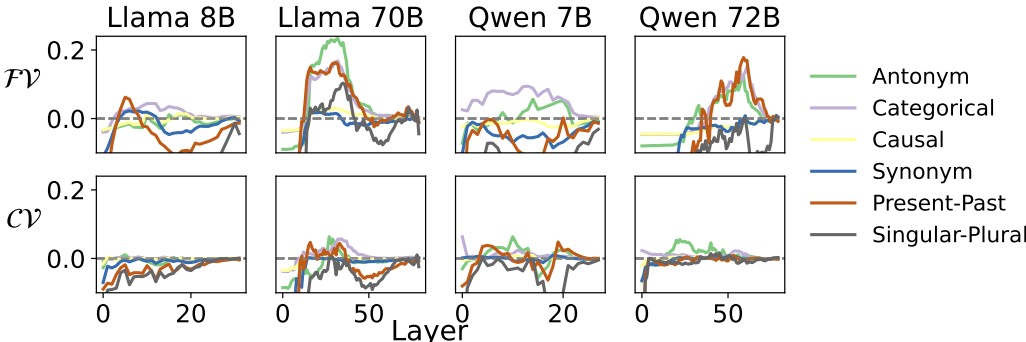

Figure 13: $\Delta$P for French translations of all the concepts. $\mathcal{FV}$s and $\mathcal{CV}$s are extracted from open-ended French prompts.

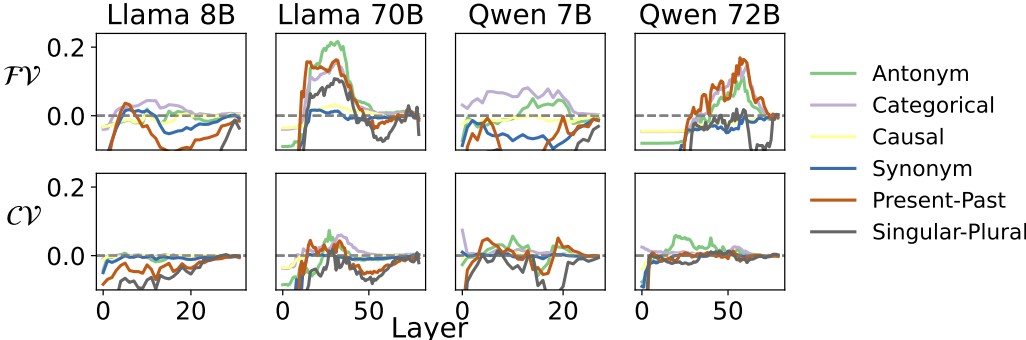

Figure 14: $\Delta$P for French translations of all the concepts. $\mathcal{FV}$s and $\mathcal{CV}$s are extracted from open-ended Spanish prompts.

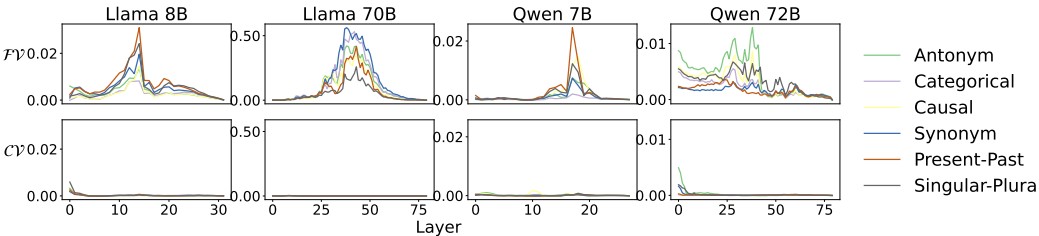

Figure 15: $\Delta$P for the opening bracket token _ (. $\mathcal{FV}$s and $\mathcal{CV}$s are extracted from multiple-choice prompts.

# H    Steering Results

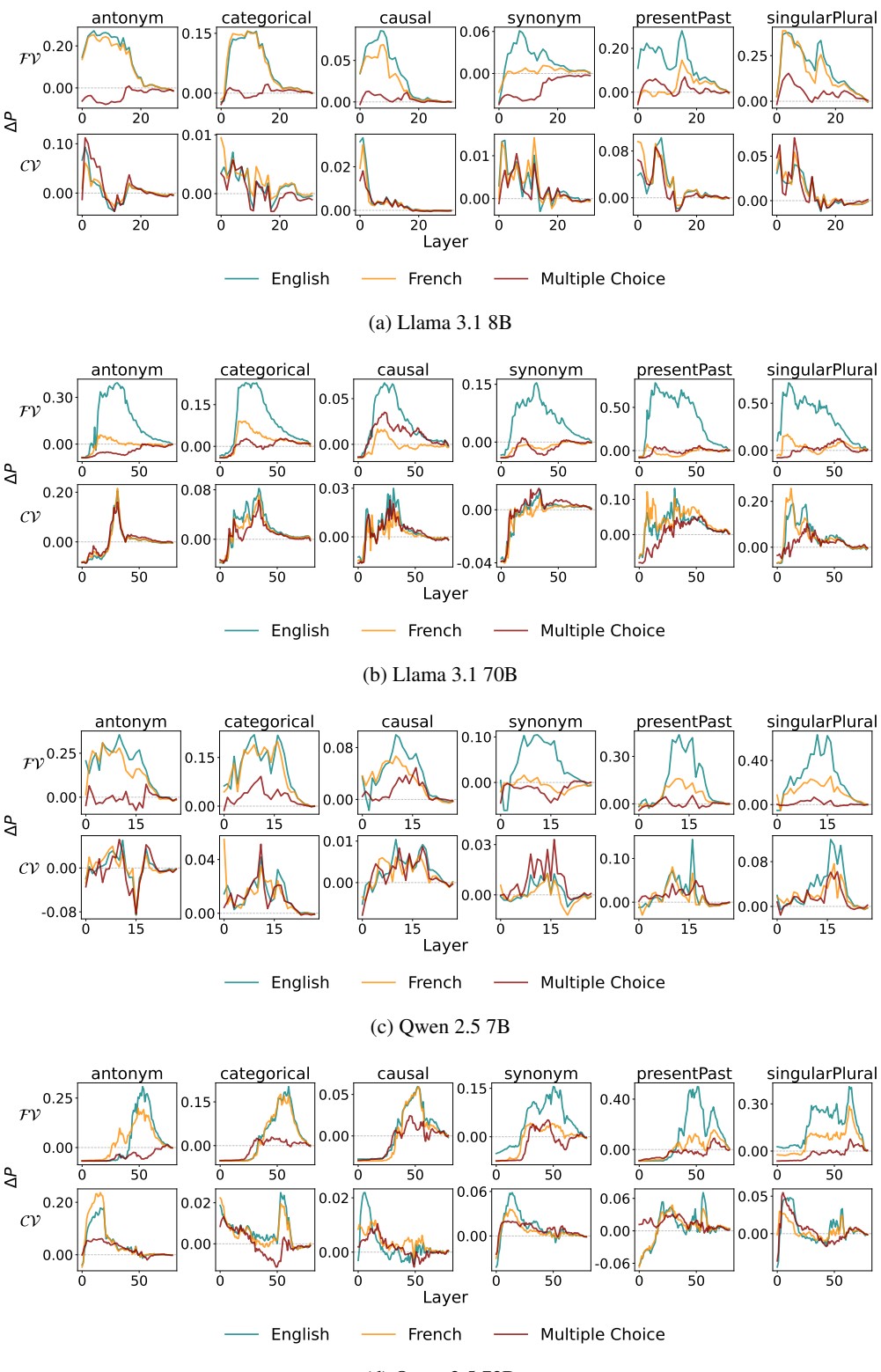

(a) Llama 3.1 8B

(b) Llama 3.1 70B

(c) Qwen 2.5 7B

(d) Qwen 2.5 72B

Figure 16: Steering effect across layers and all concepts for different models.

# I  0-SHOT STEERING RESULTS

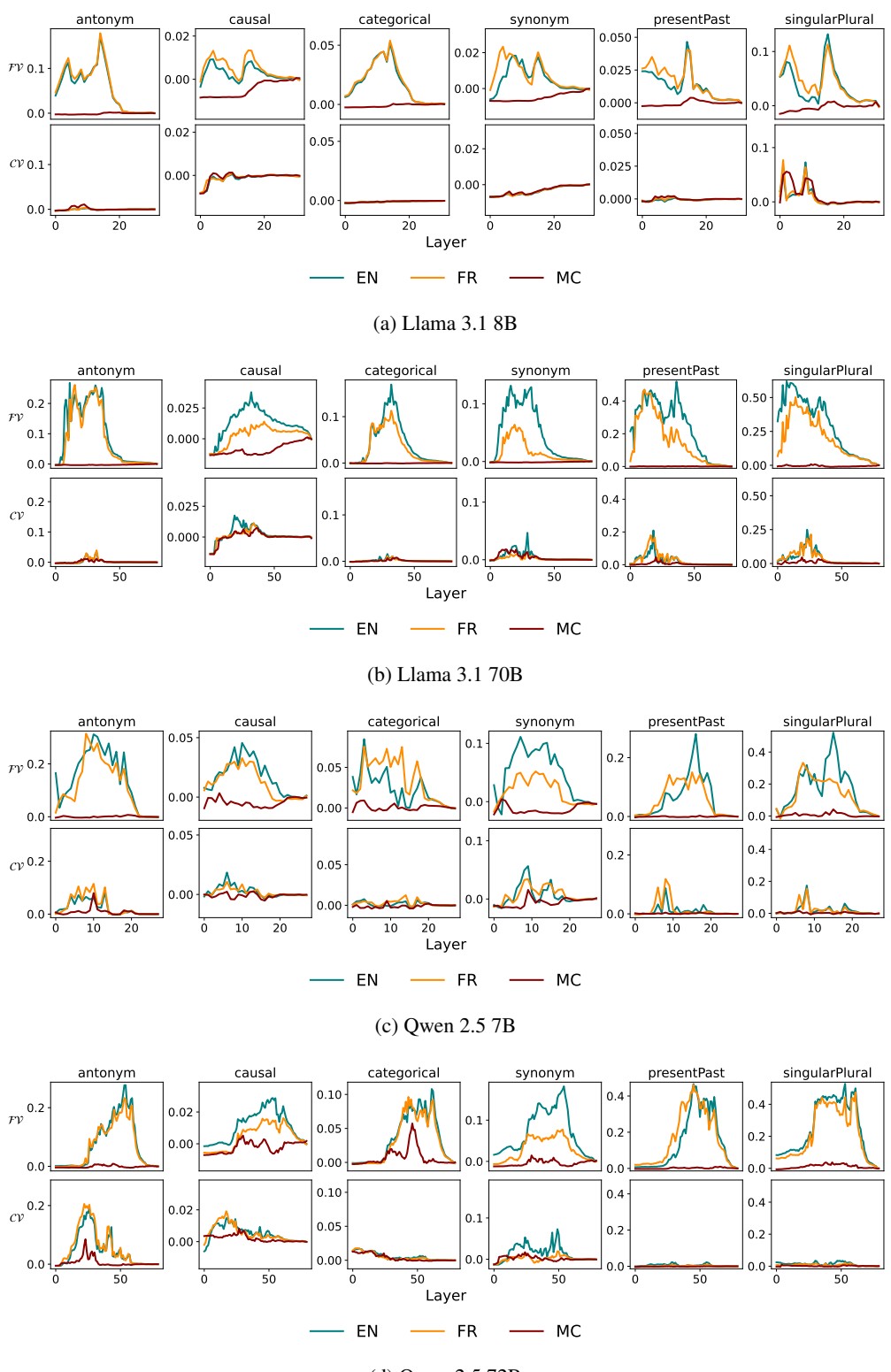

Figure 17: 0-shot steering effect across layers and all concepts for different models.

## J    QWEN 2.5 72B OUTLIER ANALYSIS

We identified anomalous CIE values for Qwen 2.5 72B in the Categorical concept across French open-ended and multiple-choice formats. As shown in Figure 18, these conditions exhibit unusually high CIE values with a bimodal distribution that deviates from the expected pattern. We excluded these two datasets from the final AIE calculations. This exclusion has minimal impact on our results: the top-5 head rankings remain identical (100% overlap), confirming that our main findings are robust to this methodological decision.

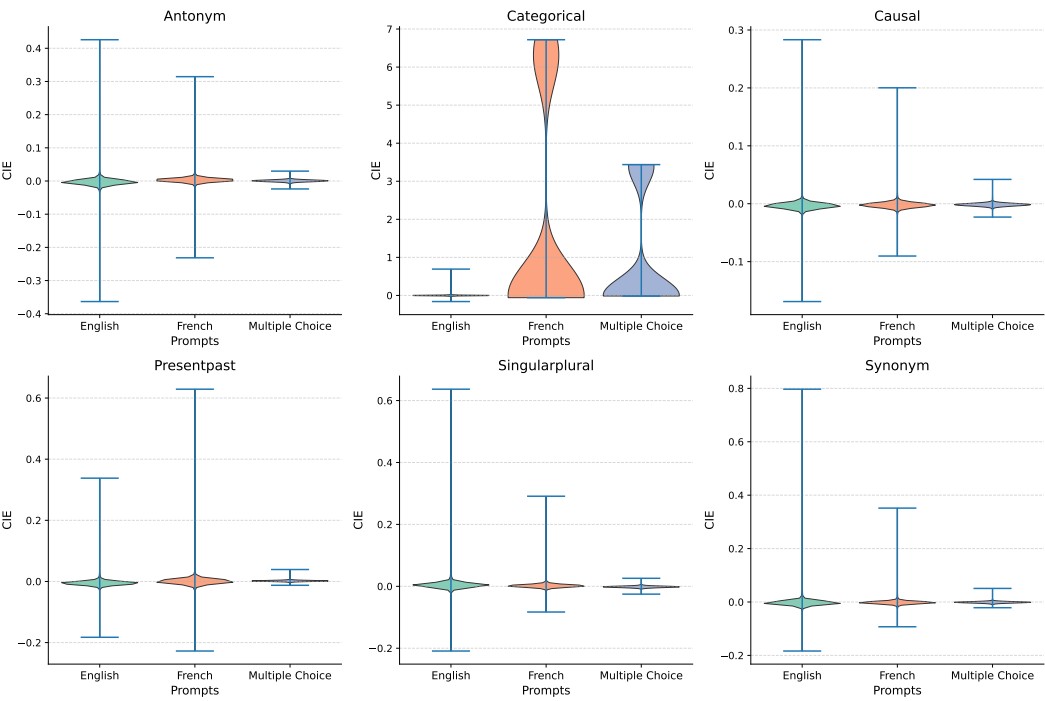

Figure 18: Violin plots of CIE for different concepts and prompts.

# K    DISENTANGLING FORMAT-SPECIFIC AND ABSTRACT REPRESENTATIONS IN FUNCTION VECTORS

Multiple-choice (MC) format involves distinct computational steps beyond open-ended generation: evaluating options, comparing candidates, and selecting among labeled alternatives (Tulchinskii et al., 2024; Wiegreffe et al., 2025). When FV heads are extracted from MC prompts, they must therefore capture both (a) the task representation (e.g., antonym, causation) and (b) the MC-specific formatting demands. We test whether partitioning out of the MC information, makes the task representation abstract, i.e., is shared across all formats?

We partition FV heads in Llama 3.1 70B into three subsets:

- **All Heads**: Top-5 heads identified by AIE computed over all input formats (standard FV selection, Eq. 5)
- **Common Heads**: Heads that appear in the top-10 heads for *all three* input formats independently. (3 heads).
- **Unique MC Heads**: Heads that appear in the top-10 heads for MC format only. (6 heads)

If abstract task representations exist in FVs independent of format, we would expect them to reside primarily in the **common heads** that are causally important across all formats. We then computed similarity matrices and RSA scores for each head subset.

We see that within the MC format, common heads cluster by concept (Figure 19), but the representations are nearly orthogonal between open-ended and multiple-choice prompts for the same concept (Figure 20).

It is still possible that these heads could encode both abstract task in open-ended and format features in MC prompts in superposition. However, under the Linear Representation Hypothesis (Park et al., 2024), a shared abstract concept should occupy a consistent linear subspace detectable via cosine similarity. The observed orthogonality across formats implies that any abstract representation is not linearly accessible in a format-invariant way. This suggests that FVs encode tasks at a lower level of abstraction (i.e., 'antonym in MC format') rather than a shared, format-independent concept.



Figure 19: *Similarity matrices for MC prompts only.* Each panel shows the similarity matrix for a different subset of AIE-selected heads computed over MC prompts only (7 concepts × 50 prompts each). Common heads show stronger concept clustering than unique MC heads (albeit with large similarity between concepts due to the shared MC structure). Full FVs show a very similar representational structure to the common heads. Model: Llama 3.1 70B.

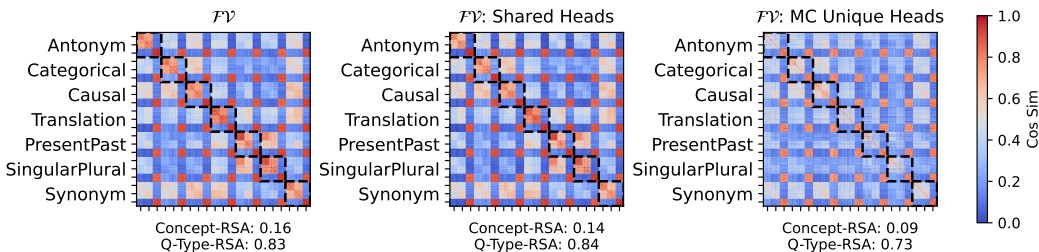

Figure 20: *Similarity matrices for different FV head subsets across all formats.* Same as Figure 19 but computed over all input formats (7 concepts × 3 formats × 50 prompts). Within the same concept the representations are nearly orthogonal between open-ended and multiple-choice prompts. Model: Llama 3.1 70B.

## L STEERING RESULTS (ACCURACY)

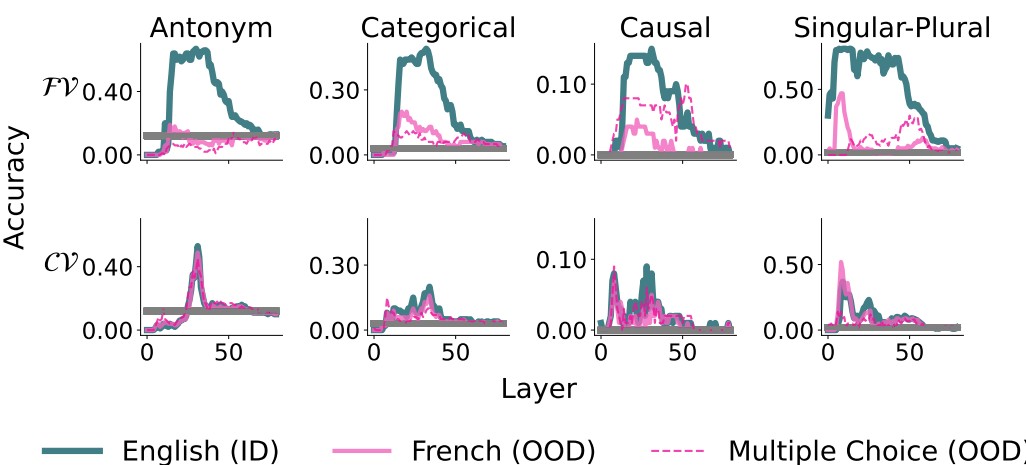

Figure 21: *Steering effect across layers (Accuracy).* We inject $\mathcal{CV}$s and $\mathcal{FV}$s into Llama-3.1-70B and plot the Top-1 accuracy for four representative concepts (columns). The grey horizontal line indicates the accuracy of the unsteered model. See Figure 7 for $\Delta P$ results.

## M MULTIPLE-CHOICE FORMAT WITH WORDS AS OUTPUT

Example prompt:

```
Instruction: Q: spherical A: ?
unconstitutional
flat
demand
healthy
Response: flat

Instruction: Q: unveil A: ?
optional
mild
sturdy
conceal
Response:
```

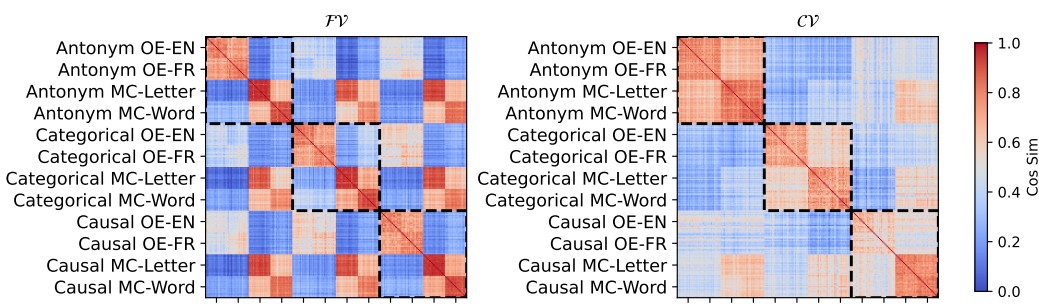

Figure 22: Similarity matrices for $\mathcal{FV}$s and $\mathcal{CV}$s with the inclusion of MC prompts where the model is expected to produce a word instead of a letter. **Takeaway**: Unlike $\mathcal{CV}$s, $\mathcal{FV}$ MC representations still cluster due to the input format, therefore MC cluster effect is not due to the model producing words/letters. Model: Llama 3.1 70B.

# N    AMBIGUOUSICL IN DIFFERENT LANGUAGES

In the original AmbiguousICL implementation, the concepts are presented in English and the translations are presented in French.

Here, we test the effect of presenting the concepts in a different language to determine if $\mathcal{CV}$s are tied to a specific surface form (e.g., "English Antonym"). Specifically, we present the concepts in Spanish and interleave them with Spanish-to-English translations.

Example antonym prompt:

```
Q: final
A: inicial

Q: inmaduro
A: madura

Q: norte
A: sur

Q: descendiente
A: descendant

Q: probablemente
A: probable

Q: vivo
A:
```

The expected response is the Spanish antonym 'vivo' → 'muerto'. Therefore, the ID vectors are extracted from open-ended Spanish antonym prompts and the OOD vectors are extracted from open-ended English and multiple-choice prompts.

In Figure 23, we see that the steering effect trends are similar to the original implementation (although the absolute performance is lower for both $\mathcal{CV}$s and $\mathcal{FV}$s). In Figure 24, we also see that the KL divergence is lower for the $\mathcal{CV}$s compared to the $\mathcal{FV}$s. Crucially, the $\mathcal{CV}$s (extracted from English) steer the model to produce the **Spanish antonym** (the contextually appropriate response), rather than the English antonym or the English translation. This demonstrates that $\mathcal{CV}$s encode the abstract 'Antonym' concept rather than 'English Antonym'. Mechanistically, this suggests that the $\mathcal{CV}$ amplifies the task probability ("do an antonym") while relying on the prompt's existing context to determine the surface form ("in Spanish"), rather than injecting language-specific content.

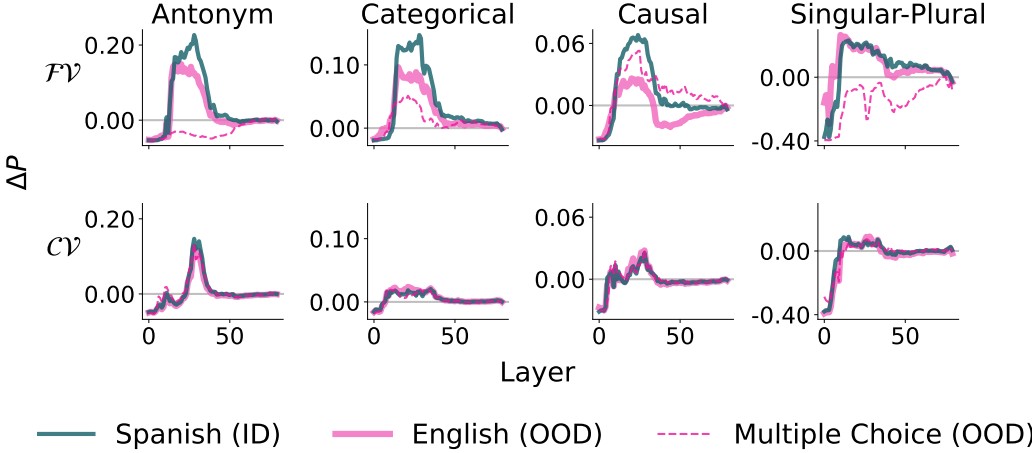

Figure 23: Steering effect across layers and all concepts for different languages.

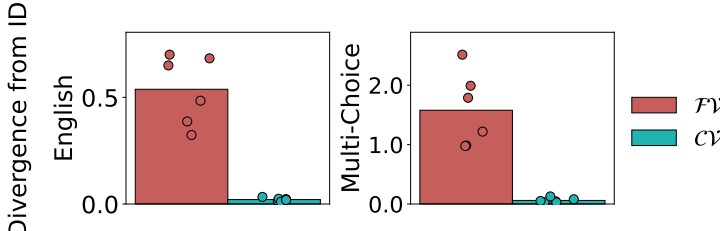

Figure 24: KL divergence between ID and OOD for different languages. Model: Llama 3.1 70B.

## O    CROSS-FORMAT ACTIVATION PATCHING

To determine whether the dissociation between causal (FV) and invariant (CV) heads stems from the activation patching setup—specifically, whether patching within the same format biases results toward format-specific heads—we conducted cross-format activation patching across all 6 combinations of input formats. In this experiment, we extracted activations from clean prompts in one format (e.g., Open-Ended English) and patched them into a corrupted run of a different format (e.g., Multiple-Choice).

We found that cross-format patching consistently identified a subset of the original Function Vector heads (e.g., in Llama-3.1-70B, head L31H18 always appears in the top-5 for both within-format and cross-format patching). It did not identify Concept Vector heads. This result shows that FVs are the primary causal mechanism for the task across all formats, despite their representations being format-specific. CVs, while representationally invariant, do not appear to causally drive the model's behavior in these tasks.

| Source Format | Target Format | Max AIE | Overlap w/ FV (top-5) | Overlap w/ CV (top-5) |
|---|---|---|---|---|
| OE-ENG | OE-FR | 0.22 | 4 | 0 |
| OE-ENG | MC | 0.02 | 2 | 0 |
| OE-FR | MC | 0.01 | 1 | 0 |
| MC | OE-ENG | 0.23 | 3 | 0 |
| OE-FR | OE-ENG | 0.41 | 4 | 0 |
| MC | OE-FR | 0.12 | 4 | 0 |

Table 4: *Cross-format Activation Patching Results (Llama 3.1 70B).* We show the max Average Indirect Effect (AIE) and the number of overlapping heads with the standard Function Vectors (FV) and Concept Vectors (CV) (top-5 heads). The patching source refers to the format from which activations were extracted, and the target refers to the corrupted prompt format into which they were patched. FV heads are consistently identified across formats (specifically L31H18 and L35H57), while CV heads are not.

