# OpenReview forum: "Causality ≠ Invariance: Function and Concept Vectors in LLMs"
_ICLR.cc/2026/Conference — ICLR 2026 Poster_

### Official Review · Reviewer_yZrA · 2025-10-17

**Soundness:** 2
**Presentation:** 4
**Contribution:** 1
**Rating:** 2
**Confidence:** 4

**Summary:**

This paper introduces an RSA-based approach to identify *Concept Vector Heads*, attention heads in LMs that capture an abstract representation of the task. The authors compare Concept Heads with Function Vector heads identified in [Todd et al, 2023](https://arxiv.org/pdf/2310.15213) and show that Concept Heads better capture the abstract task representation whereas Function vector heads tend to be more susceptible to different input formats.

**Strengths:**

A simple idea presented nicely.

**Weaknesses:**

This paper treats MCQ, more generally selecting from a list of options meeting some criteria, as trivial and treats it as a artifact of the input format. I am not sure if that is a valid assumption, See more in Question 1.

**Questions:**

1. The authors treat the MCQ format as trivial and an artifact of the input format. However, I think that MCQ or more generally selecting from a list of options meeting some criteria itself is a non-trivial task that may trigger different reasoning strategies in the LM, which are investigated in [Wiegreffe et al, 2024](https://arxiv.org/abs/2407.15018) and [Tulchinskii et al, 2024](https://arxiv.org/abs/2410.02343). Could the authors comment on this?

   1.1. When Function Vector heads are being used to capture the task representation in the MCQ format you are actually asking them to capture two things: **(a)** the abstract task representation (e.g., antonym, singular-plural, etc) and **(b)** the MCQ-formatting task (select a/b/c/d). The results presented in the paper suggest that FV heads are capturning more of (b) than (a). However, I wouldn't assume this to be a failure of FV heads as they are still capturing *a* task representation, just not the one you want them to capture. And the CV heads were identified *intentionally* to capture (a) better than (b), unlike FV heads which do not get any such signals. So, of course the CV heads will perform better when evaluated on (a). This is not a fair comparison in my opinion.

2. The authors reported probability difference as a performance metric for steerability. In my opinion, it makes more sense to report the accuracy (i.e., can CV or FV *cause* the LM to output the target answer). Although this is a harder metric, I think accuracy captures more the causal role of such abstract representations in LMs computation and was also reported in the FV paper [Todd et al, 2023](https://arxiv.org/pdf/2310.15213).

    2.1. In Figure 7, the scales in the y-axis are not aligned, neither vertically nor horizontally. The authors should at least consider aligning the y-axis scales vertically to make it easier to compare the trends across CV and FV.


### Typos and Minor Suggestions
* Replace "Open-ended" with "Open-ended ICL". Or, at least clarify when you first introduce the term.
* Line 068: *"... FVs extracted from different input formats (open-ended vs. multiple-choice) are nearly orthogonal (cosine similarity = **0.9**)"*. I think there is a typo here. Cosine similarity is a measure of alignment, so 0.9 indicates that they are very aligned, not orthogonal. Your Figure 4 also support that this is a typo.

---

> ### Author Response · Authors · 2025-11-22
>
> We thank the reviewer for your valuable feedback. We have updated our submission to address your points and answer your questions below:
>
> > CV heads were identified intentionally to capture the abstract task representation better than MCQ-formatting task, unlike FV heads which do not get any such signals.
>
> FVs in our study do receive MCQ signals. We perform activation patching on all three formats (OE-ENG, OE-FR, MCQ), and AIE scores are averaged across both concepts and formats (§2.1.2, §2.1.3, Eq. 2). FVs are not selected solely on open-ended tasks. We added a note to clarify this (L156).
>
> > When Function Vector heads are being used to capture the task representation in the MCQ format you are actually asking them to capture two things: (a) the abstract task representation (e.g., antonym, singular-plural, etc) and (b) the MCQ-formatting task (select a/b/c/d).
>
> We added an analysis (Appendix K) partitioning top-10 AIE heads in Llama 3.1 70B into: (1) heads shared across all formats (N=3), and (2) MCQ-only heads (N=6). If abstract representations existed in FVs, we would expect them in the shared heads. Figure 19 shows that for MCQ prompts, common heads do cluster by concept (RSA = 0.45). However, Figure 20 shows these representations are nearly orthogonal to open-ended ones. Full FVs showed a similar pattern to the common-heads-derived FVs.
>
> This suggests FVs encode "antonym in MCQ format" rather than format-invariant "antonym" (as CVs do). We have also added this finding to the new “Layers of abstraction” section in the Discussion (see our comments to reviewer Y7v1).
>
> > I think that MCQ or more generally selecting from a list of options meeting some criteria itself is a non-trivial task that may trigger different reasoning strategies in the LM [...] Could the authors comment on this?
>
> While we agree the mechanisms for open-ended and MCQ generation cannot be identical, it is possible—and indeed, the central question—whether FVs capture the same abstract task representation across formats. We argue that this is not obvious a priori. FVs could have been seen as encoding concepts at a higher level of abstraction as (1) they represent the causal mechanisms behind ICL; (2) abstract representations have been posited by cognitive science as necessary for analogical reasoning (see our response to Reviewer Y7v1 and updated Introduction); and (3) prior work has viewed FVs as encoding something latent about ICL tasks (see Introduction). Therefore, we think one could hypothesize a single circuit that computes the abstract concept and then executes it. We find that the mechanism is disjoint: Execution and abstraction are not just different roles but are mediated by nearly non-overlapping sets of attention heads.
>
> > CV heads were identified intentionally to capture (a) better than (b) [...] So, of course the CV heads will perform better when evaluated on (a). This is not a fair comparison in my opinion.
>
> CV heads were indeed identified intentionally to capture (a), however we view this asymmetry as a scientific finding, not a confound. AP selects for causal effect, RSA for invariance. The question is whether these properties co-localize. This empirical result implies that format-invariant representations are not the primary drivers of causal ICL effects.
>
> In our previous writeup, we might have made this comparison seem unfair by being overly critical of FVs, whereas our intention was to demonstrate that causality ≠ invariance. We updated the title to "Causality ≠ Invariance: Function and Concept Vectors in LLMs" (instead of “vs”) and added text to the Discussion (L484) clarifying that CVs are not proposed as competitors to FVs, but as distinct mechanisms.
>
> > The authors reported probability difference as a performance metric for steerability. In my opinion, it makes more sense to report the accuracy (i.e., can CV or FV cause the LM to output the target answer).
>
> We report probability differences as it shows behavioural effects in more detail. If a particular intervention shows a consistent probability boost of expected tokens, it demonstrates its causal power even if not top-1. However, as we believe many readers will share the reviewer’s interest, we now include Top-1 accuracy in Figure 21 (Appendix; and point to it on L335), which shows trends similar to the probability metrics.
>
> > In Figure 7, the scales in the y-axis are not aligned, neither vertically nor horizontally.
>
> Thank you for the suggestion! We aligned the y-axes for each column in Figure 7.
>
> >Replace "Open-ended" with "Open-ended ICL". Or, at least clarify when you first introduce the term.
>
> We updated the lines 135-138 to clarify this.
>
> > Line 068: "... FVs extracted from different input formats (open-ended vs. multiple-choice) are nearly orthogonal (cosine similarity = 0.9)". I think there is a typo here.
>
> Thank you. We removed the cosine similarity value to avoid confusion.

---

> ### Comment · Reviewer_yZrA · 2025-11-23
>
> > "FVs in our study do receive MCQ signals ..."
>
> But you select CV heads specifically to retain higher representational similarity across formats. Which suggests preferential treatment and not a fair comparison. That was my main concern.
>
>
> > Appendix K: Line 1150: "If abstract task representations exist in FVs independent of format, we would expect them to reside primarily in the common heads that are causally important across all formats ..."
>
> That is a strong assertion. The heads might take on multiple roles (superposition): encoding abstract task representation in open-ended format and encoding MCQ-formatting task in MCQ format. Thereby, Appendix K results do not necessarily contradict my point.
>
> > "While we agree the mechanisms for open-ended and MCQ generation cannot be identical, it is possible ...
>
> Thanks for your comment. You mention that *"We find that the mechanism is disjoint"*. This is indeed very plausible but I don't see any systematic ablation or causal intervention studies in your paper to justify this claim.
>
> > "CV heads were indeed identified intentionally to capture (a) ..."
>
> Thanks for the clarification. While localizing FVs, have you considered patching across formats (open-ended to MCQ and vice versa) to locate the modules encoding abstract task representation?
>
>
> > "We report probability differences as it shows behavioural effects in more detail. ..."
>
> Thanks for adding the results in Figure 21. You should've added legends so that the reader can easily interpret the lines. Consider aligning the y-axis vertically here as well.
>
> **CVs do seem to capture task encoding that remain slightly more robust across OOD formats (Figures 7 and 21). However, apart from the Antonym task, I am not sure if this gain is significant enough to justify the core claims of this paper. *I invite feedback from other reviewers on this.***
>
>
> > "We aligned the y-axes for each column in Figure 7. ..."
>
> Thanks!
>
> > "We removed the cosine similarity value to avoid confusion. ..."
>
> What was the actual value then? Fig 4 (left) is reporting RSA across Question Types, right?

---

> ### Author Response · Authors · 2025-11-27
>
> > Thanks for the clarification. While localizing FVs, have you considered patching across formats (open-ended to MCQ and vice versa) to locate the modules encoding abstract task representation?
>
> Thank you for this excellent suggestion. We have now performed cross-format activation patching (e.g., patching clean OE activations into corrupted MC prompts and vice versa) for ‘antonym’ and ‘categorical’ concepts in Llama-3.1-70B (Appendix O). We will complete the analysis for all concepts by the camera-ready deadline.
>
> Table 4 shows that cross-format patching consistently identifies a subset of the original Function Vector (FV) heads, and **not** CV heads. This further strengthens the finding that causal heads identified using AP are different from the invariant heads identified with RSA.
>
> We updated the manuscript to incorporate this new finding: Specifically:
>
> 1. in Section 2.2.2 “Function & Concept Vectors are Composed of Different Attention Heads” we added a paragraph summarizing the results (L276-280)
>
> 2. in the Discussion (“Layers of abstraction”) we mention it as an additional argument that FVs do capture some abstract information (L471-L473)
>
> 3. we added Appendix O with full experimental details
>
> Since readers might wonder why CV heads have causal effects in the AmbiguousICL task and not in the cross-format patching, we also added a discussion where we argue that CVs can only exert causal power when the concept is already present in the prompt (as is the case in AmbiguousIC) but cannot instantiate it “out of nothing” (e.g., patching experiments, or zero-shot) (L496-L500).
>
> > But you select CV heads specifically to retain higher representational similarity across formats. Which suggests preferential treatment and not a fair comparison.
>
> Thank you for highlighting this concern. We believe that with the new cross-format patching results the comparison is now fairer since both methods receive signals across formats. We hope it strengthens the reviewer’s confidence that the results highlight a legitimate dissociation, and not just a method artifact.
>
> > CVs do seem to capture task encoding that remain slightly more robust across OOD formats (Figures 7 and 21). However, apart from the Antonym task, I am not sure if this gain is significant enough to justify the core claims of this paper.
>
> We’d like to emphasize that our steering experiments were primarily diagnostic - intended to (1) confirm CVs can causally influence the model and (2) characterize FV/CV behaviour OOD (e.g. that FVs carry format artifacts) - and not focused on absolute performance (L374-376). Our core contribution is the identification of the dissociation invariance/causality dissociation.
>
> Regarding the significance: while absolute probability increases are modest, Figure 8 (KL Divergence) demonstrates that CVs induce significantly more consistent distributional shifts across formats than FVs. This supports our claim that CVs generalize better, even if they are less effective than FVs (see our contribution section; L97-98).
>
> > Thanks for your comment. You mention that "We find that the mechanism is disjoint". This is indeed very plausible but I don't see any systematic ablation or causal intervention studies in your paper to justify this claim.
>
> We agree that “mechanism is disjoint” implies a stronger claim than our data supports. To avoid overclaiming our findings we changed our contribution from “Mechanistic separation” to “CV and FV heads are disjoint.” (L94).
>
> > That is a strong assertion. The heads might take on multiple roles (superposition): encoding abstract task representation in open-ended format and encoding MCQ-formatting task in MCQ format. Thereby, Appendix K results do not necessarily contradict my point.
>
> We agree that superposition is possible. However, we have updated Appendix K (Lines 1154-1159) to clarify our point using the Linear Representation Hypothesis: if a shared abstract concept representation existed within these FV heads (e.g. a "concept direction"), it should occupy a consistent linear subspace detectable via cosine similarity or RSA. The fact that FV representations for the same concept are nearly orthogonal across formats implies that no such shared linear subspace exists. In contrast, CVs do exhibit this shared subspace.
>
> > What was the actual value then? Fig 4 (left) is reporting RSA across Question Types, right?
>
> The cosine similarity was 0.13. And yes, Figure 4 (left) reports RSA across Question Types.
>
> > Thanks for adding the results in Figure 21. You should've added legends so that the reader can easily interpret the lines.
> Consider aligning the y-axis vertically here as well.
>
> Done! We added legends and aligned the y-axes.

---

> > ### Comment · Reviewer_yZrA · 2025-11-28
> >
> > I thank the authors for being patient with me and answering my questions.
> >
> > Thanks for adding Appendix O. I found the statement below quite fascinating:
> >
> > > This result shows that FVs are the primary causal mechanism for the task across all formats, despite their representations being format-specific. CVs, while representationally invariant, do not appear to causally drive the model’s behavior in these tasks.
> >
> > I will leave one last question to the authors (which you don't have to answer): if we cannot causally manipulate the LM's behavior, how can we be sure that the CVs are indeed encoding the abstract task representation? They might be capturing some other nuanced concept (or absence of certain concepts) that simply correlates with the task, right?
> >
> > I do find the idea of localizing modules with RSA super interesting, and the presentation is superb for the most part of the paper. I am just not sure how much I should trust what concept is being captured without strong causal validation. I will encourage the authors to explore this in future work.
> >
> > I increase my rating to 4. And I wish the authors good luck with their future work!

---

### Official Review · Reviewer_eth4 · 2025-10-26

**Soundness:** 4
**Presentation:** 4
**Contribution:** 3
**Rating:** 8
**Confidence:** 4

**Summary:**

The paper investigates whether Function Vectors (FVs) are input-invariant: whether the performance with FVs vary across different input formats. They find that across different inputs, the representations are unstable. They instead introduce Concept Vectors (CVs) which are optimized via Representational Similarity Analysis (RSA). Their results show that CVs perform better out-of-domain than FVs do and that CVs and FVs are distinctly different mechanisms. Though the general performance of CVs are lower, the analysis is done well and the motivation for CVs make them a compelling source of future investigation in interpretability.

**Strengths:**

1. The figures in the paper are well made and looks very good.
2. The takeaways are clear and marked in each figure, which makes it very easy to read.
3. Overall, the paper from start to end is also simple to read.
4. The question is answered with a simple approach of RSA, and the authors do extensive analysis over in-domain and out-of-domain comparisons with both FVs and CVs.

**Weaknesses:**

1. I'd like to see the experiments with more prompts than just open-ended types and MC -- for instance, adding errors to the input prompt or small punctuation differences, if the claim is that CVs are input invariant. It's already good that there are experiments over multiple choice format too, but it would also be helpful to extend this towards formats that are "incorrect".
2. The analysis is done well to contrast in what situations FVs or CVs might be preferable. But the results also do leave the reader wondering why the heads chosen for CVs are better OOD than FVs and why CV performance may be lower than FV performance.

**Questions:**

1. Fig4, do you have some intuition about why the performance of CVs decrease as the number of K heads increases, in the "Concept" category?

---

> ### Author Response · Authors · 2025-11-22
>
> We thank the reviewer for your positive review and are glad you found the paper clear and compelling for future interpretability research. We address your questions below.
>
> > I'd like to see the experiments with more prompts than just open-ended types and MC -- for instance, adding errors to the input prompt or small punctuation differences, if the claim is that CVs are input invariant. It's already good that there are experiments over multiple choice format too, but it would also be helpful to extend this towards formats that are "incorrect".
>
> Great suggestion. We are currently running these experiments (adding errors to the input prompts) and will report results in the next few days. In the meantime you might be interested in Appendix Figure 22 which shows that CVs remain invariant even when the MC format is modified to output words instead of letters.
>
> > The analysis is done well to contrast in what situations FVs or CVs might be preferable. But the results also do leave the reader wondering why the heads chosen for CVs are better OOD than FVs and why CV performance may be lower than FV performance.
>
> CVs generalize better OOD because they are explicitly optimized for cross-format stability, effectively isolating abstract conceptual subspaces. Therefore, if a CV performs well ID, then in OOD settings it is likely that they will perform similarly, since the actual vectors are also similar. FVs likely outperform CVs in-distribution because they encode both the abstract concept and helpful low-level format signals, providing a stronger, albeit specific, steering effect.
>
> > Fig4, do you have some intuition about why the performance of CVs decrease as the number of K heads increases, in the "Concept" category?
>
> This occurs because only a small number of heads are strongly concept-aligned. Increasing $K$ forces the inclusion of heads that lack robust abstract representations, effectively adding noise to the vector.

---

> > ### Comment · Reviewer_eth4 · 2025-11-26
> >
> > Thanks for the response. I maintain my score -- good luck to the authors.

---

### Official Review · Reviewer_Y7v1 · 2025-10-27

**Soundness:** 3
**Presentation:** 3
**Contribution:** 2
**Rating:** 4
**Confidence:** 4

**Summary:**

This paper investigates whether LLMs contain representations of concepts that are robust to prompt formatting. It revisits function vectors [Todd et al.], which are implicit representations of functions constructed from attention heads. Using RSA, they find that function vector representations in MCQ settings differ from those in open-ended settings. They propose Concept Vectors, which can steer the model’s output to be invariant to the source prompt format (producing the same output even under language or structure changes).

The main contributions seem to be twofold: (1) a critique that function vectors are not invariant to prompt format, and (2) that there exist attention heads that produce concept representations that are invariant to prompt format which are separate from function vector heads.

**Strengths:**

- The experiments were clearly explained, and figures communicate key results nicely.
- The idea that the prompt form and underlying concept being tested can be separated is interesting and deserves further study.
- Robustness of methods to prompting styles/rephrasings is an important problem that others would be interested in learning about
- The paper acknowledges existing limitations of the study (e.g., concept vectors don’t steer as strongly in-distribution)

**Weaknesses:**

- The motivation of the paper (do LLMs represent concepts abstractly across surface forms?) currently feels disconnected from the experiments and results of the paper. I think it comes from the fact that this paper implicitly takes the view that a “concept” is a relation or function between two entities. Because “concept” is an overloaded term, it may be worthwhile to include an explanation/discussion of why this definition of “concept” was chosen, since it seems central to the motivation of the paper. For example, another common framing of “concept” is that of multi-token entities [Meng et al, Nanda et al], but this is very different from your view.

- The tasks used for evaluation in the paper are very simple, and because of this, the contributions appear to be mostly conceptual. This is fine by itself, though choosing stronger baselines (e.g. prompting, fine-tuning) and more complex tasks to evaluate the proposed concept vectors method would certainly strengthen the paper in terms of methodological contributions.

- The steering results in section 3.1 are somewhat mixed. In-distribution, concept vectors steer behavior somewhat poorly compared to function vectors (there are no other baselines presented). And while concept vectors do indeed seem to be invariant to the prompt format and steer towards the same answer in English under OOD formats, it’s unclear to me whether this is actually desirable behavior. Can you explain your intuition here as to why we'd want the English answer? Isn’t the idea behind in-context learning that the model should adapt based on the input? In this case, the equivariant behavior of the function vector (i.e. varies with changes in the source prompt) seems maybe more reasonable/desirable in some cases (e.g. if our goal is trying to understand ICL). Or even a composition of the two tasks (e.g. the French antonym). It seems like there is adequate evidence presented to claim that function vectors (FVs) are not exactly "invariant" to prompt format, but a discussion of “invariance” vs “equivariance” might be helpful to include because the behavior of FVs in the OOD setting seems somewhat akin to “equivariance” based on the description/examples in the paper you've provided.

___
- Todd et al. [Function Vectors in Large Language Models](https://openreview.net/forum?id=AwyxtyMwaG)

- Meng et al. [Locating and Editing Factual Associations in GPT](https://arxiv.org/pdf/2202.05262)

- Nanda et al. [Fact Finding: Attempting to Reverse Engineer Factual Recall](https://www.alignmentforum.org/posts/iGuwZTHWb6DFY3sKB/fact-finding-attempting-to-reverse-engineer-factual-recall)

- Xiong et al. [Everything Everywhere all at once: LLMs can In-Context Learn Multiple Tasks in Superposition](https://arxiv.org/pdf/2410.05603)

- Davidson et al. [Do different prompting methods yield a common task representation in language models?](https://arxiv.org/pdf/2505.12075)

**Questions:**

- A listed contribution is that concept vector heads “encode concepts at a higher level of abstraction than FV heads” - can you elaborate on what is meant by this? (Line 86)
- In Figure 3, left, it appears that the MCQ function vectors are all similar to each other. Do you think this is because they are steering the model to output a letter? Perhaps they are picking up on MCQ “task/concept” rather than the “concept” being specifically tested (e.g. antonyms). If you have the model output the answer instead of a letter in the MCQ setting and recompute the similarity matrices for function vectors, do they become concept-specific again?
- If function vectors encode both concept and format, have you tried disentangling function vectors further to extract either the concept or the format from them separately?
- The claim that "invariance and causality" are implemented separately by the model is a bit strange to me. I understand the part of the argument: attention heads for the concept vectors and attention heads for function vectors have little to no overlap. Is there more you're trying to say here beyond that? If so, I'm not sure what the evidence is that supports further claims

Other Notes:
- Related to your ambiguousICL setting, you may be interested in [Xiong et al]’s setup of task superposition, which multi-task behavior seems like a natural consequence of cross-entropy loss.
- You may also be interested in [Davidson et al], who study instruction prompt version of function vectors. They find mostly disjoint attention heads are responsible for the same task when specified via few-shot or instruction prompts.
- From your results it seems like function vectors are more “equivariant” than “invariant”, i.e., the output when steering tends to change along with the input content compared to the concept vectors. The idea that equivariance to prompting style (function vectors) and invariance to prompting style (concept vectors) are implemented separately is interesting (extrapolating from your finding that the heads are mostly disjoint). I wonder if this can tell us something about the nature of in-context learning and adaptive computation in LLMs? Perhaps it is best viewed as a composition of several mechanisms, some of which are equivariant to the context and others which are invariant for certain concepts. Curious if you have thoughts here if you have any.


- Minor Typos:
    - Figure 6, Line 296, Line 412: “Ambigous” -> Ambiguous
    - Line 852: “mulitple” -> multiple

---

> ### Author Response · Authors · 2025-11-22
>
> We thank the reviewer for your valuable feedback. We have updated our submission to address your points below:
>
> > The motivation of the paper (do LLMs represent concepts abstractly across surface forms?) currently feels disconnected from the experiments and results of the paper. [...] this paper implicitly takes the view that a “concept” is a relation or function between two entities. [...]
>
> We agree that "concept" is overloaded. We revised the paper to explicitly focus on relational concepts (mappings between entities, e.g., antonyms), as these underpin analogical reasoning in cognitive science, distinguishing them from factual associations or multi-token entities.
>
> We have updated the manuscript to ground this framing:
>
> (1) **Introduction:** We now explicitly define "relational concepts" and ask: do the abstract representations hypothesized to support analogical reasoning actually drive ICL? We link this to our finding that abstract concepts exist (CVs) but are distinct from causal mechanisms (FVs).
>
> (2) **Discussion:** Added "Analogies and abstract representation" section connecting findings to Hill’s et al. (2019) framework.
>
> > The tasks used for evaluation in the paper are very simple, and because of this, the contributions appear to be mostly conceptual. This is fine by itself, though choosing stronger baselines (e.g. prompting, fine-tuning) and more complex tasks [...] would certainly strengthen the paper [..]
>
> Thank you for voicing this concern. We adopted the task regime of Todd et al. (2024) to enable a clean comparison, while adding more abstract concepts (categorical, causal). Regarding baselines: our steering experiments were diagnostic—intended to (1) confirm CVs causally influence the model and (2) characterize FV/CV divergence OOD—rather than to propose a competitive control method. Thus, we prioritized probing representational behavior over benchmarking.
>
> > [...] In-distribution, concept vectors steer behavior somewhat poorly compared to function vectors [...] And while concept vectors do [...] steer towards the same answer in English under OOD formats, [...] Can you explain your intuition here as to why we'd want the English answer? [...]
>
> We agree that FVs' *equivariant* behavior is often desirable for practical ICL. However, since we seek abstract representations, the desirable behavior is working consistently regardless of input format (capturing just the concept). FVs producing French antonyms from French prompts implies encoding format details, not just the concept.
>
> We now clarified that AmbiguousICL is diagnostic and emphasised that the key finding is consistency, not absolute performance (L328-330; L399-400).
>
> > A listed contribution is that concept vector heads “encode concepts at a higher level of abstraction than FV heads” - can you elaborate [...]
>
> By this we mean encoding relational structure (e.g., "antonym") while discarding surface details (e.g., "English", "MC format"). FVs conflate concept with form (orthogonal across formats, but generalize within format), while CVs cluster by concept regardless of format, therefore the former encoding “antonym in MC format” and the latter “antonym”.
>
> We added (1) "Layers of abstraction" section to the **Discussion** elaborating on this and (2) a note in the contributions (L92).
>
> >[...] MCQ function vectors are all similar to each other. Do you think this is because they are steering the model to output a letter?
>
> Good question! We tested this by prompting the model to output words instead of letters in the MC setting (Figure 22, Appendix M). FVs still don’t cluster within concepts across formats.
>
> >[...] have you tried disentangling function vectors further to extract either the concept or the format from them separately?
>
> We explore this in the new Appendix K, partitioning FV heads into those shared across formats vs. unique to specific formats (MC). We find that even shared heads retain strong format dependence.
>
> >The claim that "invariance and causality" are implemented separately by the model is a bit strange to me. [...]
>
> Our claim rests on three steps: (1) FV heads are causal (confirmed by AP and prior work like Yin & Steinhardt, 2025); (2) CV and FV heads are largely disjoint (Table 1); (3) therefore, format-invariant representations (CVs) exist but are distinct from the primary causal mechanisms driving ICL. This suggests abstract structure is encoded separately from the components driving task behavior.
>
> >From your results it seems like function vectors are more “equivariant” than “invariant”. [...]
>
> Thank you for this insight! We agree and have incorporated the equivariance/invariance framing into the Discussion (L475-476). We also discuss potential interactions between CVs and FVs (or lack thereof) in the Discussion (“Limitations and Future Directions”).
>
> >You may also be interested in [Davidson et al] [...]
>
> Thank you! We included this paper in our Discussion L522.
>
> > Minor Typos:
>
> Corrected, thank you.

---

> ### Comment · Reviewer_Y7v1 · 2025-11-24
>
> Thank you for the detailed response to my questions and concerns. I think the changes have made the claims in the paper better-scoped and stronger. Because of this I'm leaning more towards a positive score.
> However, while many of my concerns have been addressed, I have a few lingering questions that I'd like to sort out.
>
>
>  > Levels of Abstraction: (Line 469): "FVs thus operate at a lower level of abstraction (e.g., “antonym in MC format”), while CVs operate at a higher level (“antonym”), independent of surface form"
>
>  - Thank you for the clarifying note about levels of abstraction in the text. However, implicit in your definition of concept vectors, there is still a "surface form" that they conform to, which is: "antonym in English" (or "X concept in English"), meaning the invariance is not just for the concept, but the concept in English. This is fine, but is something that should be discussed as a limitation of the interpretation.  For example, the "invariant" could just as easily have been "X concept in French" or "X concept in Arabic", (i.e. do all the same experiments, but see if you can find heads that perform the concept into french) but it's unclear whether you'd get the same heads that represent the invariant "concept" compared to the concept vector heads already identified in the analysis for English.
>
>
>  > **AmbiguousICL Setting:**
>
>  - Another question I have is related to the AmbiguousICL setting: have you tried the same experiment without the initial English examples (maybe replacing them with more french or spanish or something else)? While the concept vectors are extracted from various contexts, it seems like presenting examples that matches CV's implicit surface form gives it a stronger prior to "work". If they can't perform the task without the english priors, it may tell us about the limits of their invariance.

---

> ### Author Response · Authors · 2025-11-25
>
> Thank you for your continued engagement and we are glad that you find the current version of the manuscript stronger.
>
> To respond to your questions, we’d like to clarify that CV heads were identified using prompts in both English and French (as well as multiple-choice format). They are not "antonym in English" representations.
>
> Specifically:
> - Our RSA procedure (§2.1.4) computed similarity matrices over all three formats simultaneously: English open-ended (OE-EN), French open-ended (OE-FR), and multiple-choice (MC). We specify this in §2.1.2 (section “Input Formats” and further).
> - The design matrix marked pairs as sharing the same concept "regardless of the input format" (L193), which includes languages.
> - CV heads were selected precisely because they encode the same concept across both English and French (and MC).
>
> It is possible that we misunderstood your point - if that’s the case could you clarify what you mean by the ‘surface form’ of CVs is English?

---

> > ### Comment · Reviewer_Y7v1 · 2025-11-25
> >
> > Thank you for the clarification. I was focused more on the output of CVs from the AmbiguousICL experiment - they seem to be implicitly biased towards being in English. I am still hung up on why getting the English output in all settings of AmbiguousICL using the CV is desirable/expected. If the priors were changed in the AmbiguousICL setting to not be English antonyms, but Spanish antonyms mixed with French ones, does the CV still output the English version? This is why, I suppose it makes me think the CV has captured "English antonym" as opposed to just "antonym".

---

> > > ### Author Response · Authors · 2025-11-27
> > >
> > > Thank you for clarifying your concern about English priors. We reran the AmbiguousICL experiment using Spanish antonym (and other concepts) examples and Spanish-English translations. We found that the Concept Vectors consistently steered the model to produce Spanish antonyms (e.g., vivo $\to$ muerto) from all extraction sources (Spanish [ID], English [OOD] and MC [OOD]). This confirms that the vector encodes the abstract "Antonym" relation rather than a language-specific "English Antonym" representation.
> > >
> > > We also clarify why the language of the target concept examples is desirable: since the model infers the target language (Spanish, or English in the original experiment) from the prompt, the abstract Concept Vector serves only to shift the task prior (from Translation to Antonym) without overriding the language representation. We have included these results and clarification in the new Appendix N.

---

> > > > ### Comment · Reviewer_Y7v1 · 2025-11-28
> > > >
> > > > Thank you for the additional experiments and clarification. They are helpful to strengthen my confidence in the current claims and scope of the findings in the paper.

---

### Official Review · Reviewer_mXJu · 2025-10-28

**Soundness:** 3
**Presentation:** 3
**Contribution:** 2
**Rating:** 6
**Confidence:** 2

**Summary:**

This paper distinguishes two types of interpretable vector representations in LLMs: Function Vectors (FVs) derived via activation patching and Concept Vectors (CVs) derived via representational similarity analysis (RSA). Through systematic experiments across seven conceptual relations and three input formats (open-ended English, multilingual, and multiple-choice), the authors show that FVs capture causal features that drive in-distribution performance but are sensitive to surface format, while CVs encode more abstract, format-invariant concept representations. The two sets of attention heads occupy similar layers but are largely disjoint, suggesting distinct mechanisms for causal effectiveness and conceptual abstraction.

**Strengths:**

Well-designed methodology combining activation-based and representational analyses; broad and carefully controlled experimental setup (languages, formats, concepts); clear empirical evidence supporting the causal vs. invariant distinction; insightful visualization and analyses of head overlap and cross-format transfer; overall a solid and novel contribution to mechanistic interpretability of ICL.

**Weaknesses:**

- It is not surprising that the function vector is different from concept vector. Function vector is responsible in "doing this task", where concept vector is tailored for "summarizing the topic of this task". One is execution (based on low-level semantics such as current formats) and one is abstraction.
- A detailed study of the mechanism - how the two sets of heads cooperate with each other semantically - is crucial but currently absent.
- Several related work might be of insteast and warrent a detailed discussion:

Yang et al. Unifying Attention Heads and Task Vectors via Hidden State Geometry in In-Context Learning

Bu et al. Provable In-Context Vector Arithmetic via Retrieving Task Concepts.

Han et al. Emergence and Effectiveness of Task Vectors in In-Context Learning: An Encoder Decoder Perspective

- 1050 prompts might be insufficient.

**Questions:**

What suggestions would you provide to deep learning theoretical literature to model the function-vector and concept vector? Especially, for Bu et al., what can you suggest for their theoretical modeling to consider tasks beyond factual recall? Would there be any better algebratic manner to handle theoretical modeling of in-context learning with those vectors based on your findings, in the consideration of broader task types?

---

> ### Author Response · Authors · 2025-11-22
>
> We thank the reviewer for their feedback and are glad you found the paper a solid and novel contribution to mechanistic interpretability of ICL. We address your questions below.
>
> > It is not surprising that the function vector is different from concept vector. Function vector is responsible in "doing this task", where concept vector is tailored for "summarizing the topic of this task". One is execution (based on low-level semantics such as current formats) and one is abstraction.
>
> We argue that this is not obvious a priori. FVs could have been seen as encoding concepts at a higher level of abstraction as (1) they represent the causal mechanisms behind ICL; (2) abstract representations have been posited by cognitive science as necessary for analogical reasoning (see our response to Reviewer Y7v1; updated Introduction; and new “Analogies and abstract representation” Discussion section); and (3) prior work has viewed FVs as encoding something latent about ICL tasks (see Introduction). Therefore, we think one could hypothesize a single circuit that computes the abstract concept and then executes it. We find that the mechanism is disjoint: Execution and abstraction are not just different roles but are mediated by nearly non-overlapping sets of attention heads. We update our framing in the Introduction to clarify why we think it’s a reasonable assumption to make that “execution” and “abstraction” could co-localize.
>
> > A detailed study of the mechanism - how the two sets of heads cooperate with each other semantically - is crucial but currently absent.
>
> We agree that understanding the mechanistic interaction (if one exists) between these two sets of heads is a critical next step. In this work, our primary contribution and scope is establishing the existence and separation of these two distinct types of representations. We now provide two detailed hypotheses in the Discussion (“Limitations and Future Directions” section) to be explored in future work - (1) CVs and FVs interact during inference as detection/execution mechanisms (2) CVs and FVs do not interact during inference; CVs are simply a backup circuit.
>
> > Several related work might be of insteast and warrent a detailed discussion:
>
> We thank the reviewer for the suggestions. We have now added discussion of:
>
> Han et al. (2025): Encoder/decoder perspective on task vectors, which we connect to our CV/FV distinction as potential detection/execution mechanisms (Discussion, “Limitations and Future Directions” section, L511).
>
> Bu et al. (2024): We added a discussion on the implication of our findings to theoretical models of ICL (L480-485).
>
> > What suggestions would you provide to deep learning theoretical literature to model the function-vector and concept vector? Especially, for Bu et al., what can you suggest for their theoretical modeling to consider tasks beyond factual recall? Would there be any better algebratic manner to handle theoretical modeling of in-context learning with those vectors based on your findings, in the consideration of broader task types?
>
> While we are not theoretical modelers, our findings suggest two key constraints for frameworks like Bu et al. (2025):
>
> *Format conditionality*: Bu et al. model ICL as retrieving a single task vector $a_\theta^f(T)$ for a function $f$. However, we find that for the same function, vectors extracted from different formats are nearly orthogonal. Thus, theoretical models should define the task vector as $a_\theta(f, \phi)$, conditioned on surface format $\phi$. This suggests that models converge to multiple format-specific basins rather than a single global abstract minimum.
>
> *Orthogonality of abstraction and execution*: We find that FVs and CVs are nearly orthogonal to each other within the residual stream. This implies that the "task representation" partitions into two distinct subspaces one for the abstract concept and one for the format-specific execution rather than a single unified vector.
>
> We also discuss this in L480-485.

---

### Author Response · Authors · 2025-11-22

We thank the reviewers for their thoughtful feedback. We have updated our manuscript to address your comments. We respond to each individual reviewer in separate responses.

We invite the reviewers to assess the submission and consider increasing scores if your questions have been addressed.

Key improvements include:
- **Refined Framing**: We updated the Introduction and Discussion to focus on relational concepts and their connection to analogical reasoning.
- **Revised Title and Scope**: We changed the title to "Causality ≠ Invariance: Function **and** Concept Vectors in LLMs" (instead of vs.) and added text to clarify that our goal is to dissociate invariance (CVs) from causality (FVs) rather than present them as competitors. We also refined the text to clarify that our causal analyses are meant to be diagnostic, rather than focusing on absolute performance.
- **Layers of Abstraction**: We added a new section to the Discussion defining abstraction as encoding relational structure while discarding surface details. We clarify that CVs operate at a higher level of abstraction (concept only), while FVs operate at a lower level (concept + format).
- **New Analyses**:
    * Appendix K shows that FVs remain format-conditional even when restricting analysis to heads shared across formats.
    * Appendix L shows Top-1 accuracy plots.
    * Appendix M shows that FVs remain format-conditional even when changing MC questions to output a word instead of a letter.
- **Future Hypotheses**: We added a discussion on potential interactions between CVs and FVs (as detection vs. execution mechanisms) and implications for theoretical ICL models.

---

### Author Response · Authors · 2025-12-03

We thank the reviewers for their valuable feedback. Their suggestions helped us improve the overall contribution of the paper.

Since our last general comment, we have made the following additions and clarifications:

- Appendix O: We evaluate cross-format activation patching and find that it consistently identifies a subset of the original FV heads, but not CV heads, directly addressing reviewer yZrA’s recommendation and concerns.
- Appendix N: As suggested by reviewer Y7v1, we extended the AmbiguousICL experiments to multiple languages and observe consistent trends.

We believe these additions, together with the changes described in our earlier general comment, address the main concerns raised in the reviews and clarify the contributions of our work.

---

### Meta-Review · Area_Chair_nwCG · 2025-12-07

**Summary:**

This study proposes a conceptual distinction between "function vectors" (FVs), which encode information about a task in certain formatting-dependent contexts, and "concept vectors" (CVs), which encode information about a task in a more abstract manner (that is less invariant to input format). It is observed that vectors derived via patching experiments tend to be format-dependent, whereas vectors derived via RSA are far less so. It is also observed that these two types of vectors are largely disjoint.

Reviewers appreciated the conceptual straightforwardness of the main ideas, and were largely positive on the merits of these ideas for researchers working in this area. Concerns primarily focused on (1) the scope of experiments being limited to narrow and simple task settings, (2) somewhat mixed empirical results when directly comparing FVs to CVs, and (3) potentially unfair comparisons between FVs and CVs.

**Reviewer Concerns:**

1. This was partially addressed via additional follow-up analyses, but reviewers did not seem to find it a major hurdle to acceptance in the first place, as the point of the paper is primarily to demonstrate the existence of a distinction between FVs and CVs. It would have made the study stronger to fully address this with more complex task settings, but the evidence is sufficiently strong as-is to make an interesting distinction that could impact future work.

2. Discussions on this point with Reviewer Y7v1 yielded stronger evidence, but follow-up points made by Reviewer eth4 suggest that this was not fully resolved.

3. Further discussions with eth4 and analyses added to the manuscript did a good job of addressing this point. While there is still some uncertainty around the causal strength of the difference between FVs and CVs, there no longer appears to be a concern that the comparison is scientifically invalid.

**Reviewer Scores:**

Reviewers Y7v1 and eth4 have explicitly stated that they had wanted to increase their scores in response to the discussion and revisions. Reviewer mXJu did not respond, but the second weakness they raised was left to future work, while the third weakness was relatively minor. However, the response to the first weakness raised by mXJu was specific and convincing, in my opinion, so this could have gone either way; I lean toward either the same score or a slight increase.

---

### Decision · Program_Chairs · 2026-01-26

Accept (Poster)